# Highly efficient and selective extraction of gold by reduced graphene oxide

Fei Li[1,5], Jiuyi Zhu[1,5], Pengzhan Sun [2], Mingrui Zhang[1], Zhenqing Li[1], Dingxin Xu[1], Xinyu Gong[1], Xiaolong Zou [1], A. K. Geim [1,2]✉, Yang Su [1]✉ & Hui-Ming Cheng [3,4]✉

Materials capable of extracting gold from complex sources, especially electronic waste (e-waste), are needed for gold resource sustainability and effective e-waste recycling. However, it remains challenging to achieve high extraction capacity and precise selectivity if only a trace amount of gold is present along with other metallic elements . Here we report an approach based on reduced graphene oxide (rGO) which provides an ultrahigh capacity and selective extraction of gold ions present in ppm concentrations (>1000 mg of gold per gram of rGO at 1 ppm). The excellent gold extraction performance is accounted to the graphene areas and oxidized regions of rGO. The graphene areas spontaneously reduce gold ions to metallic gold, and the oxidized regions allow good dispersibility of the rGO material so that efficient adsorption and reduction of gold ions at the graphene areas can be realized. By controlling the protonation of the oxidized regions of rGO, gold can be extracted exclusively, without contamination by the other 14 co-existing elements typically present in e-waste. These findings are further exploited to demonstrate recycling gold from real-world e-waste with good scalability and economic viability, as exemplified by using rGO membranes in a continuous flow-through process.

Electronic waste (e-waste) is the world's fastest-growing solid waste and poses risks to the environment and human health. Less than 20% of e-waste has currently been recycled, primarily for a lack of technologies with sufficient efficiency and economic viability to recover valuable elements within it[1–5]. Gold is the most valuable part of e-waste, and its efficient extraction can turn this recycling challenge into a profitable business[6–8]. Activated carbon is widely used for gold extraction but has significant drawbacks, including low extraction capacity, poor selectivity, and high energy and resource intensity[9–11]. There is a strong demand to develop novel gold extraction materials with higher extraction capacity and

selectivity. Various novel gold adsorbents have been explored recently. These gold adsorbents could be categorized into two sets, one is nanoporous materials, for example, metal-organic framework[12,13], covalent organic polymer[4,14], and porous aromatic framework[15]. The gold extraction behaviour of these materials is mainly contributed by the immobilization of gold ions with the intrinsic porosity and the chemical reduction of the gold ion by the added functional groups. The other set of gold adsorbents, for example, two-dimensional molybdenum disulphide[16–18], amyloid[19,20], cyclodextrin[21] and diamide[22], instead of relying on the porosity of the adsorbents, their efficient gold extraction is

[1]Tsinghua-Berkeley Shenzhen Institute & Institute of Materials Research, Tsinghua Shenzhen International Graduate School, Tsinghua University, Shenzhen 518055, P. R. China. [2]School of Physics & Astronomy, University of Manchester, Manchester M13 9PL, United Kingdom. [3]Faculty of Materials Science and Engineering / Institute of Technology for Carbon Neutrality, Shenzhen Institute of Advanced Technology, Chinese Academy of Sciences, Shenzhen 518055, P. R. China. [4]Shenyang National Laboratory for Materials Sciences, Institute of Metal Research, Chinese Academy of Sciences, Shenyang 110016, P. R. China. [6]These authors contributed equally: Fei Li, Jiuyi Zhu. ✉e-mail: Geim@manchester.ac.uk; Su.yang@sz.tsinghua.edu.cn; Cheng@imr.ac.cn

accounted to the chemical reduction of gold ions to Au[0] by photoreduction or the functional groups, and precipitation of gold ions with the adsorbents.

These gold adsorbents exhibit a high gold extraction capacity at a gold concentration from 500 ppm-3000 ppm[12–22], but this capacity decreases to less than 250 mg/g at a gold concentration range relevant to e-waste recycling, specifically, from ppb level to tens of ppm[4,12,19]. Furthermore, as the e-waste contains complex co-existing metal elements, the practical gold extraction process often requires good extraction selectivity to gold, so that separation of co-existing elements from the extracted gold can be avoided, achieving a high resource- and energy- efficiency for gold recycling. The existing novel gold adsorbents have demonstrated good selectivity to gold, but their adsorption to the co-existing metal elements is still unneglectable[4,12,15,16,19]. Therefore, the development of materials with high gold extraction capacity to trace the amount of gold, precise gold selectivity and economic viability remains lacking.

Here we report an exceptionally high gold extraction capacity of chemically reduced graphene oxide (rGO), reaching 1850 mg/g and 9059 mg/g when extracting gold from its 10 ppm solution at 25 °C and 60 °C, respectively, combined with an ability to extract gold at minute concentrations, down to parts per trillion and high selectivity. During extraction, rGO reduces >95% gold ions to metallic gold, avoiding elution and precipitation necessary in post-adsorption processing. Moreover, gold extraction can be done selectively, without adsorption of the other 14 elements normally present in e-waste. This, in turn, enables the recycling of copper, the second most valuable metal in e-waste. Finally, we demonstrate a highly efficient flow-through process for gold extraction using rGO membranes. Our findings show a promising venue for addressing global e-waste challenges and gold scarcity.

## Results and Discussion
### Highly efficient gold extraction by rGO

The rGO was obtained by chemical reduction of commercial, mass-produced graphene oxide (GO) nanosheets using ascorbic acid as a reductant (Methods). The resulting rGO nanosheets were repeatedly washed with deionized water, until a stable colloid was formed, and then the rGO suspension was added directly into solutions containing gold ions (Fig. 1a). Considering the hydrochemical process is a flexible, low-cost, and sustainable method for e-waste recycling[23,24], and [AuCl$_4$]$^-$ is a common gold complex seen in a hydrochemical process[4,7,12], we have chosen KAuCl$_4$ solutions prepared in different concentrations (C) to examine the gold extraction capacity of rGO (Fig. 1a, b); Similar to the behaviour observed for other adsorbents[4,13,14], the extraction

capacity increased with increasing C and reached a plateau above 10 ppm (Fig. 1b). Specifically, 1 g of rGO can extract ~ 690, 1180 and 1850 mg of gold from 0.1, 1 and 10 ppm gold solutions, respectively. This extraction capacity outperformed those of novel nanoporous adsorbents at the same concentrations and, importantly, extends to solutions containing only 1 ppb of gold (Supplementary Note 1; Supplementary Fig. 1 and Supplementary Table 1). Although we could not quantify the extraction capacity for even lower C (our quantitative methods reached their detection limit), we still observed significant gold extraction down to as little as 10 ppt (Supplementary Fig. 1). This makes the reported rGO technique particularly interesting because wastewater and freshwater often contain gold with C below 10 ppb, and its efficient reclaim is challenging[12].

When measuring the extraction capacity of rGO to different pH from 2 to 11, we found the extraction capacity varied with the pH, which is not unusual and similar to many gold adsorbents[4,13–16,19–21]. As shown in Supplementary Fig. 2, the maximum gold uptake was found at pH ≈ 4. This is attributed to the fact that, at lower pH, the rGO colloid loses its stability[25,26] whereas, at basic pH, rGO undergoes deprotonation, becomes negatively charged[4] and starts repelling negatively charged [AuCl$_4$]$^-$. Both effects decrease the extraction capacity. It is important to note that the extraction capacities at the acidic and basic solution are still higher than the previously reported gold extraction capacities at the same C (Supplementary Figs. 1 and 2), and by increasing the amount of rGO, ~99% gold ions from its 10 ppm solution with the pH of 2−11 can be extracted (Methods and Fig. 1b). We also investigated the gold extraction as a function of time (Fig. 1c and Supplementary Fig. 3). Within 10 min, 1 g of rGO provided extraction of ~1010 and 325 mg of gold from 10 ppm and 100 ppb solutions, respectively (Fig. 1c), suggesting not only efficient but also rapid extraction. Interestingly, after extraction to 1 ppm gold, rGO changed its colour from black to gold within hours, suggesting almost instantaneous Au recovery (Supplementary Movie 1).

### Mechanism study of rGO's gold extraction behavior

To understand the ultrahigh gold extraction behavior of rGO, we started with the characterization of rGO after the gold extraction. We found nanoparticles on top of rGO nanosheets after 24 h of its exposure to gold solution (Fig. 2a). X-ray diffraction (XRD) showed this particulate to be metallic gold, suggesting that [AuCl$_4$]$^-$ was reduced to Au[0] (Supplementary Fig. 4). This was confirmed by X-ray photoelectron spectroscopy (XPS) analysis (Fig. 2b) which showed that >95% of the particulate was metallic Au[0] rather than [AuCl$_4$]$^-$. Thermal analysis (Supplementary Note 2; Supplementary Fig. 4) provided good agreement with the XPS results, as no endothermic peak for the

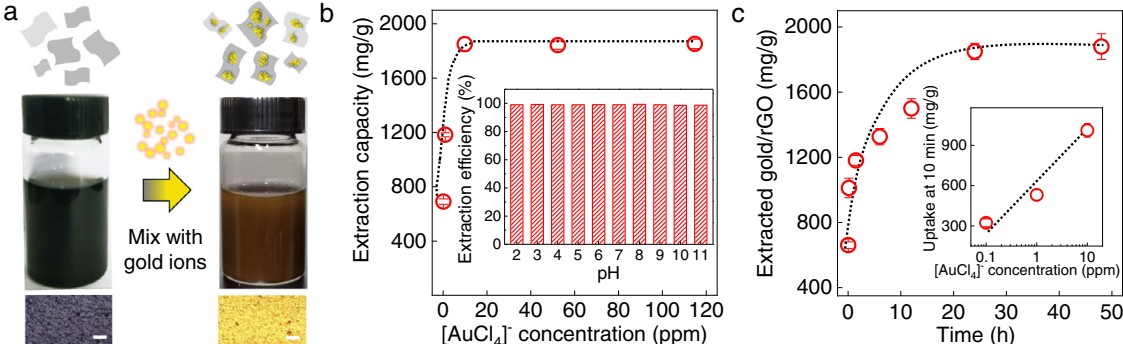

**Fig. 1 | High-efficiency extraction of trace amounts of gold. a** Schematic of the extraction process using rGO. After mixed with gold ion (10 ppm) for 12 h, rGO suspension gradually changed its color from black to brown. The bottom panels are the optical images of rGO films deposited from rGO nanosheets before (black) and after gold extraction (gold). Scale bar: 20 μm. **b** Extraction capacity as a function of gold concentration after 24 h. The inset plots the extraction efficiency measured for 10 ppm solutions at different pH. **c** Extraction capacity as a function of time for a 10 ppm gold solution. Inset: Extraction from different solutions after 10 min. Dashed lines, guides to the eye. All the experiments were performed at 25 °C. All the error bars in this figure represent the standard deviation.

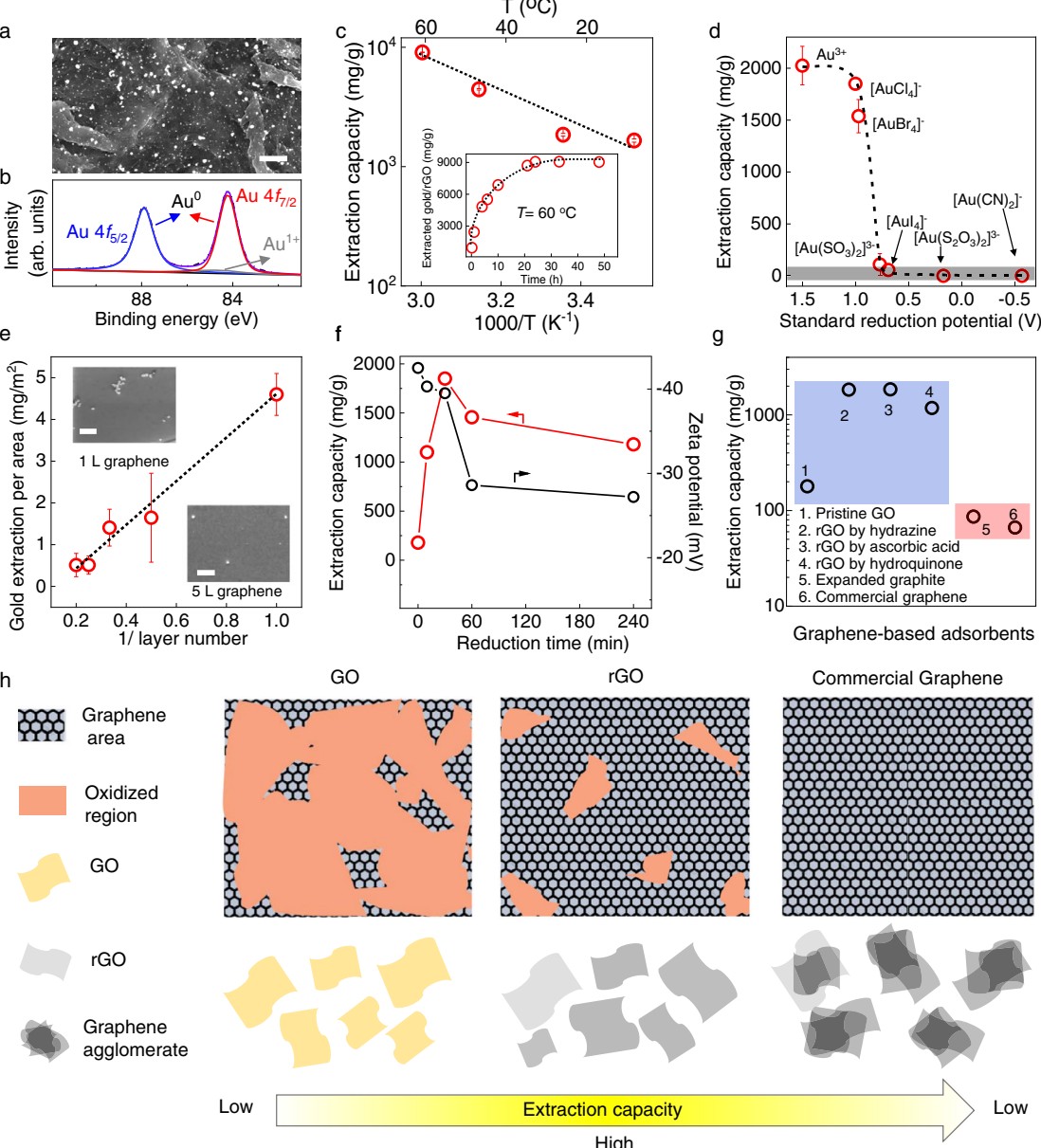

**Fig. 2 | Understanding the gold extraction mechanism. a** Scanning electron microscopy (SEM) image and **b** 4f XPS spectrum of gold nanoparticles on rGO. Black curve: raw data with the fitting envelope in purple. The deconvolved peaks for $Au^0$ and $Au^+$ are colour coded. Scale bar in **a**, 500 nm. **c** Temperature-dependent gold extraction behaviour. Inset is the extraction capacity as a function of time for a 10 ppm gold solution measured at $T = 60\,°C$. **d** Extraction capacity for gold complexes with different reduction potentials[52]. The shaded area marks extraction capacities below 100 mg/g. Dashed lines: guides to the eye. **e** Areal gold extraction capacity measured for mechanically exfoliated graphene. The error bars indicate standard deviation using several graphene crystallites. The insets are SEM images of exfoliated graphene ($N = 1$ and 5) after gold extraction. The scale bars are 500 nm. **f** Extraction capacity (measured at $C = 10$ ppm) and zeta potential of rGO versus its reduction time. **g** Extraction capacity measured for different graphene-based adsorbents. **h** The schematic of atomic structures for GO, rGO and commercial graphene, their corresponding dispersibility and extraction capacities. All the experiments except those in **c** were performed at 25 °C. All the error bars in this figure represent the standard deviation.

transformation of gold salts into $Au^0$ was found by differential scanning calorimetry (DSC). These observations suggest that a reductive adsorption mechanism dominates the extraction process and, if any adsorption of $[AuCl_4]^-$ on rGO occurs, its contribution is minor. Furthermore, we found that gold nanoparticles formed already after 2 min of the contact of rGO with gold solutions (Supplementary Note 3; Supplementary Fig. 7). Moreover, DSC analysis after 10 min showed no peak attributable to the gold salt precursor (Supplementary Fig. 8). This suggests that the reductive adsorption of gold ions happens throughout the entire extraction process, including its initial stages. In comparison to the reported novel gold adsorbents, though some showed reductive adsorption, other physi- and chemi-sorptions were also significant. That is, gold extracted by these adsorbents was a mixture of ionic and metallic gold[4,12–15,19,27–29], therefore, it would require energy- and cost-intensive post-processing, for example, elution and precipitation to desorb and reduce the ionic gold, so that a full adsorption capacity can be achieved[30,31]. In contrast, as >95% extracted Au ions were reduced to metallic gold by rGO, it allowed direct isolation of metallic gold from rGO surface without further post-adsorption processing, providing an extra advantage of rGO for gold extraction.

Because no extra reductant for gold ions was used during gold extraction, it is sensible to conclude that the observed reductive

adsorption was a redox reaction between the gold ions and rGO. We have performed gold extraction at different temperatures ($T$) to investigate the kinetics of this redox reaction. As the gold extraction capacity is a direct result of the redox reaction rate, we have plotted the extraction capacity versus $T$, and fitted with the Arrhenius equation, $\exp(-E/k_BT)$, that $E$ is the energy barrier and $k_B$ is the Boltzmann constant. This yielded an energy barrier $0.12 \pm 0.02$ eV (Fig. 2c), suggesting an activation behaviour for gold extraction, which is consistent with our DSC analysis that gold reduction by rGO is an endothermic process (Supplementary Fig. 4), the $E$ is much lower than the activation energy seen in typical chemisorption process (0.21 eV-4.3 eV)[32], supporting the observed fast and ultrahigh extraction capacity at low $C$. Furthermore, the observed temperature dependence allowed improving gold extraction capacity by increasing $T$, for example, at $T = 45$ °C and 60 °C, we observed extraction capacities ($C = 10$ ppm) of 4100 mg/g and 9059 mg/g respectively.

Next, we studied the influences of the redox pair, gold ion and rGO on the gold extraction performance. For the gold ions, we have measured rGO's adsorption capacity to gold ions with different reduction potential $E_O$ (Fig. 2d). Similar capacities were found for solutions of $Au^{3+}$, $[AuCl_4]^-$ and $[AuBr_4]^-$. Note that these complexes were most frequently used for gold extraction by novel adsorbents as well as for doping of graphene by gold-containing solutions[4,12,14,21,33,34]. This also suggested the possibility for rGO to extract gold from gold solutions containing different gold ions. As a demonstration, we found rGO reclaimed ~100% gold from 10 ppm $[AuBr_4]^-$ solution obtained by dissolving gold in the aqueous solution of N-bromosuccinimide at pH = ~8[35]. For ions with $E_O < 0.8$ eV, for example, $[Au(CN)_2]^-$ and $[Au(S_2O_3)_2]^{3-}$, rGO showed little extraction, corroborating our conclusion that the redox reaction between gold ion and rGO dictated the reported behaviour.

To study the influence of rGO on the redox reaction, we recalled the local atomic structure of rGO nanosheet (Fig. 2h), which contains large unoxidized (or reduced) graphene areas, some oxidized regions that provide stability of rGO dispersions, and sub-nanometre pinholes[36]. Given a little fraction of pinhole (~5%) on rGO[36], we only focused on the contribution of graphene area and oxidized regions to the gold extraction behaviour.

For the graphene area, in previous reports of gold ion ($Au^{3+}$ or $[AuCl_4]^-$) doped graphene grown by chemical vapour deposition (CVD), the graphene was found to chemically reduce these gold ions via a redox reaction mechanism, in which graphene donated electrons to reduce $Au^{3+}$ to $Au^0$.[37,38] Though these studies only focused on a high $C$ (≥200 ppm) and a short doping time (one to few minutes), they implied the observed reductive adsorption likely was accounted to the graphene area.

To validate that the graphene area of rGO reduced the gold ion, we have first studied the gold reduction by pristine graphene obtained by mechanical exfoliation, so that possible influence of the oxidized region can be excluded. The gold nanoparticles formed on the graphene after a few minutes and continued to grow even after many hours. Using areal extraction capacity allowed us to quantify the gold extraction capability of mechanically exfoliated graphene. The monolayer graphene showed the highest areal extraction capacity, in quantitative agreement with the weight capacity observed for rGO (Supplementary Note 4), suggesting the graphene area in rGO was responsible for ultrahigh gold extraction capacity. Rather unexpectedly, as shown in Fig. 2e, increasing the layer number of graphene ($N$), the areal gold extraction capacity, which should be independent of $N$ as only the outmost graphene layer had contact with the gold ion, and there's no gold intercalation between graphene layer, instead, decreased with $N$, indicating not only the redox potential between graphene and gold ions is essential, but also other factors are involved in the process. Further SEM analysis revealed that wrinkles and folds frequently seen for mono- and few-layer graphene tended to accumulate more gold nanoparticles than in flat areas (Supplementary

Figs. 9 and 10), suggesting a higher chemical activity for gold extraction at the warped graphene, which was in agreement with previous reported gold doped CVD graphene[37], this can be explained by our simulation (Supplementary Fig. 11) that the strain generated in the warped area of graphene, decreased the adsorption energy to gold ion and enhanced electron transfer of graphene to gold ion, both were beneficial for reductive adsorption of gold ion (Supplementary Note 4).

Further insights on the contribution of the graphene area to the gold reduction were obtained by considering the electron transfer from the graphene area to gold ions, the Raman and ultraviolet-visible (UV-Vis) spectroscopy were used to probe the electron transfer of rGO by comparing the spectra before and after gold extraction. The Raman spectrum showed that after extraction, the rGO became p-doped and showed an increased number of defects in the rGO nanosheets[39] (Supplementary Figs. 5a and 6). The UV-Vis spectroscopy revealed that the characteristic peak of rGO shifted towards that of GO (Supplementary Fig. 5b). These two pieces of evidence indicated the donation of electrons from rGO for gold reduction[40], confirming the reduction of gold ions by the graphene area of rGO. In another set of experiments, encouraged by the fact that graphene reductively adsorbed gold ions, the gold extraction capacities of commercial graphene and expanded graphite were measured. Both graphene materials were known to have well-retained graphene areas but little oxygen-containing functional group, which was also verified by the Raman analysis (Supplementary Note 5, Supplementary Fig. 12). During the extraction, they showed no dispersibility in gold solution because of their hydrophobicity, and their extraction capacities were <100 mg/g (Fig. 2g).

The low extraction capacity observed in commercial graphene suggested graphene area was not the only factor contributing to the ultrahigh extraction capacity of rGO. Thus, we studied the influence of the other atomic structure, oxidized regions of rGO, on the extraction performance. Given the synthesis of rGO was a process to remove some oxidized regions on GO (Fig. 2h), we have controlled the reduction time of GO, and XPS analysis confirmed the oxidized regions were gradually removed with the reduction time (Supplementary Note 5, Supplementary Fig. 12a). As shown in Fig. 2g, at a $C = 10$ ppm, pristine GO showed little gold extraction capacity, but the rGO obtained by 10-minute reduction exhibited an extraction capacity ~1100 mg/g, and 30-minute reduction gave the highest extraction capacity ~1850 mg/g (Fig. 2f), which could be explained by the increased graphene area in the rGO with a prolonged reduction time. Therefore, more sites were available for reductive adsorption. Further increasing the reduction time to 1 and 4 h, rGO agglomerated as validated by their decreased zeta potentials, and the capacities have decreased to ~1460 and 1180 mg/g, respectively. This can be understood by that the reduction was beneficial for the recovery of graphene areas of rGO, but a long-time reduction excessively eliminated the oxidized regions. As a result, agglomeration and a poor dispersibility of the rGO were seen, such agglomeration of rGO disabled some of the recovered graphene areas from being exposed to the aqueous gold ion and inhibited the reductive adsorption, leading to a low extraction capacity (Fig. 2h). In addition, we also have tried rGO reduced by hydrazine or hydroquinone for gold extraction. Both rGO had recovered graphene area and formed stable monolayer dispersions as confirmed by their zeta potentials (Supplementary Note 5; Supplementary Fig. 12). They exhibited high extraction capacities (~1190 and 1835 mg/g from $C = 10$ ppm, respectively) as shown in Fig. 2g, similar to the values for the rGO reduced in ascorbic acid. These experiments unambiguously showed that both the graphene area and the oxidized regions were critical for the observed high gold extraction capacity.

With the above experimental results, we proposed the mechanism for the ultrahigh gold extraction of rGO as follows, firstly, the adsorption was driven by the concentration difference ($\Delta C$) of adsorbates in its solution and on the adsorbent[41]. In our case, $\Delta C = C_{sol}^{Au\ ion} -$

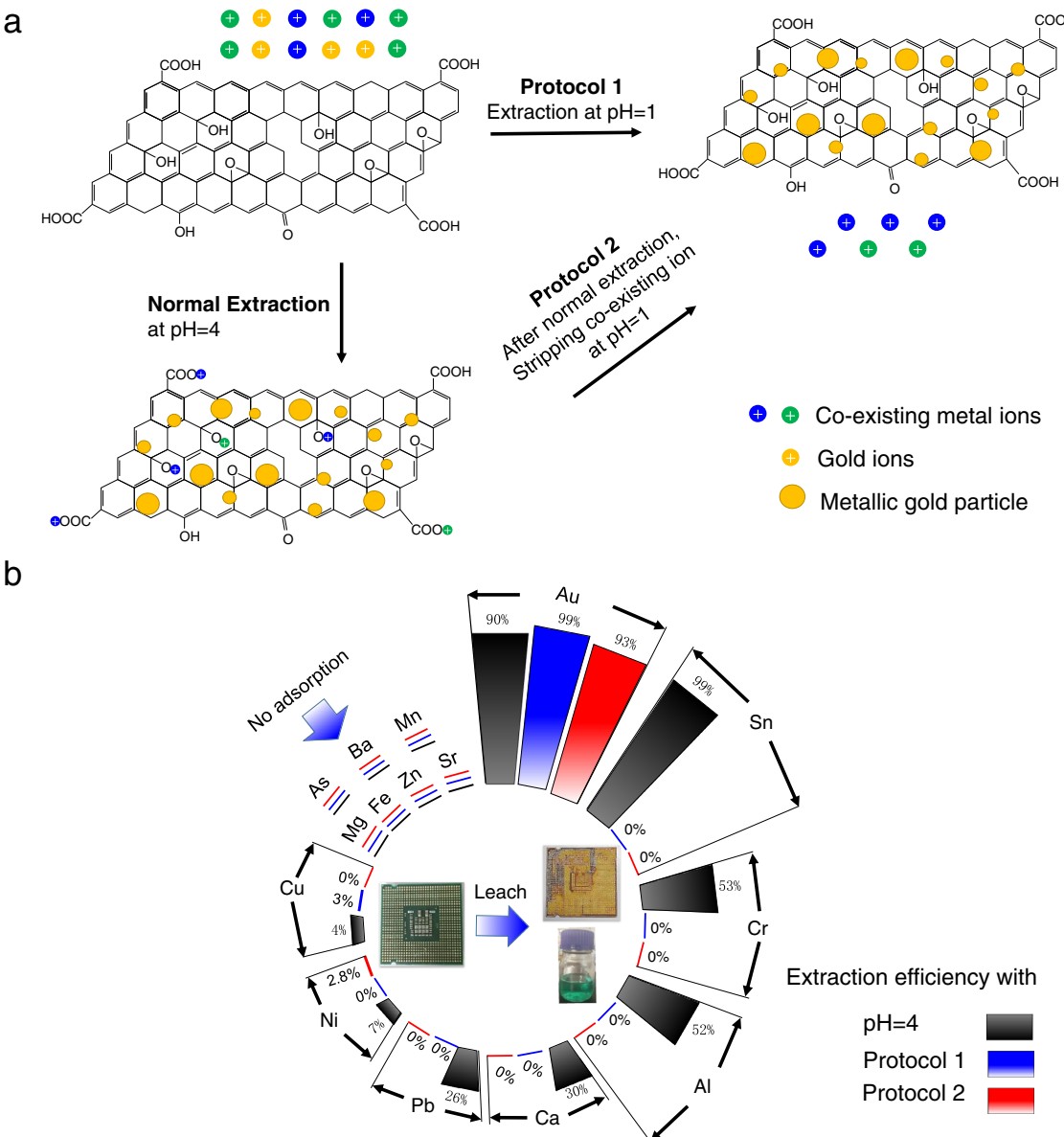

**Fig. 3 | Highly selective gold extraction. a** Schematics show two additional protocols to increase selectivity by changing pH. Co-existing ions are prevented from adsorption (protocol 1) or stripped away later, after reductive adsorption of Au (protocol 2). **b** Comparison of ion selectivity using different extraction protocols. The photos inside the circle show the discarded, leached CPU and the resulting leachate. All the experiments were performed at 25 °C.

$C_{rGO}^{Au\ ion}$, Once gold ions were adsorbed on rGO, >95% of them were reduced to $Au^0$ spontaneously. Such rapid and complete conversion of gold ion to $Au^0$, left an extremely low $C_{rGO}^{Au\ ion}$, consequently, a high $\Delta C$ was maintained during the adsorption process. Therefore, rGO overcame the fast equilibrium at low C, and showed an ultrahigh adsorption capacity at even ppb level. Secondly, the reductive adsorption was enabled by the graphene area of rGO, it donated electrons to the adsorbed gold ion, which was reduced to $Au^0$. In addition, the wrinkles and warped area of graphene prompted the adsorption of gold ions and electron transfer to gold ions during the redox reaction, further increasing the gold extraction capacity. The oxidized regions of rGO, provided a good dispersibility of rGO, therefore allowing efficient adsorption and reduction of gold ions by graphene area, leading to an ultrahigh extraction capacity (Fig. 2h).

### Realization of high selectivity for gold extraction
Based on the described understanding, we explore the possibility of extracting Au from e-waste that typically contains a variety of other

metals. Initial tests were performed using a simulated e-waste mixture containing $[AuCl_4]^-$, $Cu^{2+}$, $Ni^{2+}$ and $[PtCl_4]^{2-}$ ions. We observed ~99% gold recovery with <5 % of the other metals being extracted (Supplementary Fig. 13). Next, a discarded central processing unit (CPU) was leached with aqua regia, forming leachate containing ~88 ppb of gold ions. Our rGO colloids captured ~90% of this gold (Fig. 3b) but also took up co-existing ions, similar to the other previously used gold adsorbents[4,10,12,14]. The uptake was from ~7 to 100 % for 6 out of the 14 metals present in the CPU leachate in concentrations from sub-ppm to hundreds of ppm (Fig. 3b).

Because high selectivity is essential for viable recycling, we further improved our extraction procedures, considering that adsorption sites for gold and co-existing metal ions were likely to be different. Indeed, gold ions were reduced on graphene areas of rGO, whereas residual functional groups were reported to provide efficient adsorption of metal ions[42–44]. By changing the pH of rGO suspensions from basic to acidic, the functional groups could be (de-)protonated reversibly. Deprotonation results in more negatively charged rGO nanosheets

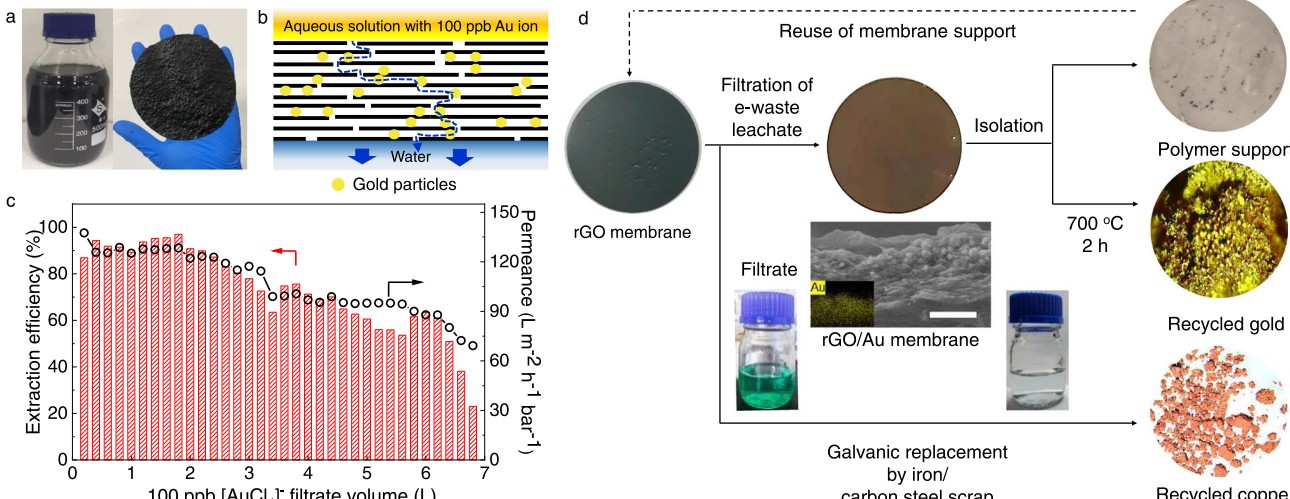

**Fig. 4 | Flow-through technology for gold extraction and recycling. a** The left and right images show photos of rGO suspension and rGO membrane of 100 cm². **b** Schematic of low-concentration gold extraction using rGO membranes. **c** Extraction efficiency and water permeance were reduced with increasing the amount of the 100 ppb Au solution filtrated through. **d** Schematic of a gold extraction process from e-waste leachates. The left, middle, and upper right images show photos of the initial rGO membrane, its state after filtration (at 50% extraction efficiency) and the polymer filter used for rGO membrane, respectively. Central panel: cross-sectional SEM micrograph of the final rGO membrane with gold nanoparticles seen accumulated between rGO nanosheets. The inset of the SEM image: EDS map of elemental gold. Scale bar, 500 nm. The container on the left is the leachate filtrate after gold extraction. The filtrate became colourless after galvanic extraction of copper (container to the right). The images presented as middle and lower circles to the right are photos of our final products: gold after burning rGO and galvanic copper, respectively. All the experiments were performed at 25 °C.

that electrostatically attract and adsorb metal ions. In contrast, protonation prevents such adsorption. We exploited this consideration to achieve gold extraction with little co-adsorption of other ions within the CPU leachate (Fig. 3a). Two protocols were developed. In the first one, gold was extracted directly from a highly acidic (pH ≈ 1) leachate (Fig. 3a). As shown in Fig. 3b, this allowed an extraction efficiency of 99.3% at gold $C$ ≈ 88 ppb, whereas <3% of Al and Cu were extracted with undetectable adsorption of the other metals. This protocol could be particularly suitable for gold extraction involving strong acids, for example, if aqua regia is used to dissolve e-waste. In the second protocol, we showed that protonation effectively strips the adsorbed co-existing ions on rGO after their adsorption. First, we exposed rGO to the CPU leachate at pH ≈ 4 for 24 h, which allowed maximum adsorption of gold. Then, pH was reduced to ~1 for an hour, which stripped off adsorbed co-existing ions (Fig. 3a). We obtained 93% gold extraction from the same CPU leachate with <2% for Ni and no other metals (Fig. 3b). Furthermore, for leachates with higher (2.65 ppm) Au concentrations and using the second protocol, noting in this case, the concentration of Cu was nearly two orders of magnitude higher than gold, still, ~99% of gold was extracted without the detectable presence of any other metal (Supplementary Fig. 15). The success of these protocols strongly supports the suggested model of site-specific ion adsorption on rGO.

Gold extraction from seawater offers an extreme challenge because of minute Au concentrations being present, believed to be <20 ppt[12,45]. To address this challenge, we prepared simulated seawater and added 10 ppt of gold to it. Using our normal extraction procedures, we estimated complete gold extraction, indicating a possibility of gold mining from oceans (Supplementary Note 6). The Au uptake can probably be increased further by developing special protocols similar to those described above (Supplementary Fig. 14).

## Continuous flow-through gold extraction

The described technology is scalable, as the process only involves mild temperature reduction, the rGO can be scalably made (Fig. 4a). Furthermore, to demonstrate its potential for scalable gold extraction, we

also developed a scheme involving a continuous flow of low-$C$ Au solutions through rGO membranes (Fig. 4b; Supplementary Note 8). Figure 4c shows the extraction performance of a membrane with a thickness of ~800 nm and 2 cm in diameter during filtration of several litres of a 100 ppb gold solution. As gold nanoparticles gradually accumulated between rGO nanosheets and started blocking the water flow (Fig. 4c), both extraction efficiency and membrane's permeance decreased. This decrease typical for any adsorption-based membrane separation[46,47] can be moderated using membranes in series. XRD analysis confirmed the presence of metallic gold within the resulting rGO membranes (Supplementary Fig. 16), their cross-sectional SEM and energy dispersive spectroscopy (EDS) imaging showed that gold was adsorbed and reduced through the entire cross-section (Fig. 4d). As an ultimate test for the technology's viability for e-waste recycling, the rGO membranes were used with real CPU leachates. The observed permeance and efficiency were close to those for pure gold solutions (Supplementary Note 8, Supplementary Fig. 17). As the final step to isolate metallic gold from our rGO, because of superior adsorption capacity and selectivity to gold, further elution and precipitation process are not required in our case. We, therefore, used the melting process which is the last step of the gold recycling process, to burn the membranes in the air at 700 °C, which left gold particulate (Fig. 4d). Its EDS analysis confirmed >95% purity with the rest being carbon, oxygen and sodium (Supplementary Fig. 16), this further proved that the adsorption of co-existing ions (both cationic and anionic ions) was little, in good agreement with the observed precise gold selectivity of rGO. We also tried regeneration of rGO by dissolving gold extracted by rGO with thiourea/HCl[16] and found regenerated rGO showed a gold extraction capacity of ~1000 mg/g ($T$=25 °C). However, considering GO we used is a mass-produced commercial product which has a price of less than 0.5 RMB per gram (Shenzhen Matterene Technology), and the high temperature for removing rGO is, nevertheless, used for gold melting purposes, therefore, direct isolation at high temperature may be a better choice than regeneration with respect to the process cost and sustainability. It is worth noting that the nearly complete Au extraction from e-waste also simplifies the sortation of other metals present in leachates. For example, after gold extraction, we used the

remaining filtrate to extract valuable copper by using its galvanic replacement with iron (Fig. 4d). The recycled copper had >95% purity (Supplementary Fig. 18).

Finally, we point out that the entire recycling process can be viable from an economic perspective. If we define the 50% extraction efficiency as the end life for rGO membranes, a 1 m$^2$ membrane (<3 g) under a 1 bar pressure is estimated to allow extraction of ~1.6 g of gold from ~20 tons of a 100 ppb leachate. As the typical cost of GO is less than 0.5 RMB per gram, and the gold price is ~300 RMB per gram, together with the observed ultra-high gold extraction capacity, precise selectivity, and scalability, though continuous efforts are required for its commercialization, rGO provides a considerable incentive for commercial recovery of gold from e-waste. Even rGO-based extraction from seawater might be viable as the process does not consume other reactants.

## Methods
### Preparation of rGO
We used commercial GO (Shenzhen Matterene Technology) synthesized by a modified Hummers' method[48]. In a typical reduction process using ascorbic acid[49], pH of a 20 mL 0.5 mg/mL GO suspension was first adjusted to ~10 by adding an ammonia solution. Under stirring at room temperature, 35 mg of ascorbic acid was added, and the resulting GO suspension was placed for the reduction in an oil bath at 95 °C for 30 min. This time was found optimal because shorter times diminished the area of unoxidized graphene, whereas longer times resulted in agglomeration of rGO nanosheets, with both effects degrading the extraction capacity (Fig. 2f). The rGO suspension was centrifuged at 12,850 g, and the sediment was re-dispersed in water by sonication at 400 W for 30 min, resulting in rGO nanosheets with an average lateral size of 100−500 nm. This washing process was repeated at least three times to remove unreacted ascorbic acid and by-products generated during the reduction. The final rGO aqueous dispersion had a concentration of 0.5 mg/ml. For hydroquinone-reduced GO, 176 mg of hydroquinone was added to 20 mL 0.5 mg/mL GO suspension[50], and the same procedure as described above for ascorbic acid was employed. For making hydrazine-reduced GO, we followed the procedures reported elsewhere[51]. We mixed the 20 mL 0.5 mg/mL GO suspension, 50 μl of ammonia solution (28 wt% in water) and 14 μl of hydrazine solution (50 wt% in water). Then the same procedure as described above for ascorbic acid was carried out.

### Extraction capacity of rGO
The capacity was measured using gold ion solutions (KAuCl$_4$) mixed with rGO suspensions to obtain mixtures with final gold concentrations of 0.1, 1, 10, 50 and 100 ppm. The gold solutions were prepared by diluting the stock solution of KAuCl$_4$ with deionized water. pH of the mixtures was adjusted to 4 by adding HCl or NaOH. The weight ratio of Au ion to rGO was kept at 2:1, and we note for 10 ppm and higher concentration, further increment of this weight ratio (Au ion to rGO) did not result in an increment in extraction capacity at $T = 25$ °C. We used extraction times from 2 min to 48 h (Fig. 1c) while shaking the mixtures. At each time point, the mixtures were filtered using a 13 mm PES membrane syringe filter with 0.22 μm pore size. Then the filtrates were analysed by inductively coupled plasma optical emission spectroscopy (ICP-OES) and inductively coupled plasma mass spectrometry (ICP-MS) to determine the adsorption capacity, $Q_e$ (mg/g). It was calculated as $Q_e = \frac{(C_0 - C_e) \times V}{m}$ where $C_o$ is the initial concentration of Au (ppm), $C_e$ is its final concentration in the filtrate (ppm), $V$ is the volume of the used suspension (L) and $m$ is the mass of dry rGO (g). The gold extraction was usually performed in the dark, but no notable differences were found under light conditions. To measure the influence of pH, we followed the same procedures as above and changed pH by adding either HCl or NaOH. To ensure reproducibility, all the measurements were repeated at least 3 times. For gold extraction by mechanically exfoliated graphene is detailed in Supplementary Note 4, and for rGO's gold extraction from different gold complexes, we followed a similar procedure as gold extraction from [AuCl$_4$]$^-$ solution, the exception was [Au(CN)$_2$]$^-$, that the extraction was performed at a pH = ~10 as [Au(CN)$_2$]$^-$ becomes unstable in acidic solution. Unless otherwise noted, all experiments were conducted at room temperature (25 °C). In order to study the influence of temperature on gold adsorption properties, extraction experiments were conducted at 10, 25, 45 and 60 °C respectively. The weight ratio of Au ion to rGO was kept at 10:1 in this case.

### Extraction efficiency
We measured extraction efficiency for 10 ppm solutions at pH of 2−11. The pH was adjusted by adding 0.5 M of HCl and 0.5 M of NaOH solutions. 10 mg of rGO were added to 20 mL of the gold solution with final gold concentration of 10 ppm. After shaking for 24 h, the mixtures were filtered and analysed by ICP-OES to determine the extraction efficiency, $R$ (%). It was calculated as $R = \frac{C_0 - C_e}{C_0} \times 100\%$.

### Extraction selectivity
In our single-metal selectivity tests, we used an aqueous solution of CuCl$_2$, NiCl$_2$ and K$_2$PtCl$_4$ to achieve 10 ppm concentrations of Cu$^{2+}$, Ni$^{2+}$ and Pt (present as [PtCl$_4$]$^{2-}$ ions). The weight ratio of each metal ion and rGO was 2:1. For selectivity measurements in the presence of several metal ions, we mixed these solutions, as well as a 10 ppm Au solution in equal volumes. Then, we followed the above-mentioned procedures for gold, repeating each experiment at least 3 times for reproducibility. All experiments were conducted at room temperature (25 °C).

### Real-world gold extraction
A discarded CPU was obtained from computer waste. To leach gold from the CPU, it was first soaked in an 8 M NaOH solution for two days to remove the protective coating on the electronic surfaces. Afterwards, the CPU was rinsed and soaked in 40 mL aqua regia at 60 °C for two days. Undissolved material was filtered and rinsed with deionized water to form leachate with $C$ ~100 ± 20 ppb. pH of the leachate was changed to 4 by adding NaOH. 3.6 mg of rGO was added to 20 mL of the final leachate and stirred for 24 h at room temperature. Then we followed the procedures described above for gold extraction measurements. In the first protocol, we adjusted the pH value of the leachate to 1, using HCl and only then performed the gold extraction. In the second protocol, after gold adsorption on rGO at pH ≈ 4, we again used HCl to reach pH ≈ 1, and the mixture was shaken for an extra hour. The amount of extracted gold was evaluated using the above procedures with at least 3 samples. All experiments were conducted at room temperature (25 °C).

### Flow-through extraction and copper recycling
A suspension containing ~1.8 mg of dry rGO was filtered through a cellulose membrane support for 12 h by vacuum filtration at 25 °C. Note that if we vacuum-filtrated for much longer times (e.g., 72 h), rGO deposits became completely dry and exhibited no permeability to water. The resulting (wet) rGO membranes were tested for their performance using a 100 ppb gold solution. The permeance and gold extraction efficiency were measured each time after 200 mL of the solution was filtered. The efficiency was calculated as $R = \frac{C_0 - C_e}{C_0} \times 100\%$. We found a trade-off between the permeance and efficiency such that thicker membranes led to lower permeance but higher uptake of gold (Supplementary Fig. 16). We chose to work with 800 nm-thick membranes (measured in the dry state) because of a good balance between permeance and uptake.

To demonstrate copper extraction from e-waste, we used the leachate left after extracting gold by filtration through rGO. Iron

particles were added directly into the filtrate for 30 min after which no further precipitation was noticeable. The reddish-brown precipitate was then collected, washed in 0.1 M HCl and then in deionised water, and vacuum dried for chemical analysis.

## Materials characterization

Metal concentrations in aqueous solutions were measured using ICP-MS (iCAP RQ) and ICP-OES (Optima 7300 DV). XRD was performed using a Bruker D8 advance diffractometer operated at 40 kV and 40 mA using Cu Kα radiation ($\lambda = 0.154$ nm). The morphology of rGO after extraction was examined by SEM (Hitachi SU8010) operated at 5 kV. DSC and TG measurements of $KAuCl_4$, rGO and the products after gold extraction were obtained using the analyser Jupiter STA 449 F3 at heating and cooling rates of 10°C/min from 20 to 800 °C. XPS measurements were conducted on spectrometer PHI-5000 Versa Probe II. Zeta potential measurements were performed using Zeta sizer Nano ZS. UV–Vis absorption spectra were recorded on a UV–Vis spectrophotometer (UV-2600, Shimadzu). Raman spectra were recorded on a Horiba Evolution HR Raman spectroscope using a 532 nm argon ion laser.

## Data availability

The data that support the plots within this paper and other findings of this study are available within the article and the Supplementary Information file, or available from the corresponding authors upon request. Source data are provided with this paper.

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

## Acknowledgements

We thank Prof. Jinping Zhao for providing us with GO samples at the start of the project, we thank Mr. Yusheng Tan and Mr. Xitao Lin for the ICP tests and Mr. Qi Yan for providing seawater, and Prof. Yuxiong Huang from Tsinghua SIGS for his insightful discussion with us. This research is supported primarily by National Key Research and Development Program of China (No. 2019YFA0705400), the Scientific Research Start-up Funds of Tsinghua SIGS (Grant No. QD2021026C to Y.S.) and Shenzhen Geim Graphene Center.

## Author contributions

Y.S., H.-M.C. and A.K.G. conceived and supervised the project, Y.S., F.L. and J.Y.Z. designed the experiments. P.Z.S. performed experiments on gold extraction by mechanically exfoliated graphene. Z.Q.L and X.L.Z. performed theoretical simulation. F.L. and J.Y.Z. performed the rest of the experiments in this report. M.R.Z., D.X.X. and X.Y.G. helped with the sample preparation. All authors analyzed the data, and F.L., Y.S. H.-M.C and A.K.G. wrote the manuscript with input from all authors.

## Competing interests

Y.S., F.L. and H.-M.C. are listed as co-inventors on two pending Chinese patent applications related to this work filed by Shenzhen International Graduate School, Tsinghua University (no. 202111087418.7 and 202210375264.X). All other authors declare no competing interests.
