## [Peer Review File · Nature Communications]

Highly efficient and selective extraction of gold by reduced graphene oxideReviewers' comments:

Reviewer #1 (Remarks to the Author):

In the present manuscript, the authors reported and demonstrated that rGO enables effective and selective Au uptake from various water matrices. Various advanced characterization techniques were employed to verify the effectiveness of rGO for the reductive adsorption of Au. Although some interesting results were present, I still believed that this study fail to provide enough new insights on the underlying working mechanism. The reported sorption capacity and selectivity were also comparable or only slightly better than few recent publications (both batch and flow-through operations). I, hence, cannot recommend its publication in Nature Communications.

My detailed comments are given below:

1. Similar reductive adsorption mechanism toward Au uptake were reported. The new insights need to be clearly illustrated and the novelty need to be justified as well.
2. The Au sorption isotherm and kinetics need to carried out systematically and fairly compared with previous reports (under normalized conditions);
3. The authors claimed that "the areal gold extraction efficiency also depend on other factors", what are these other factors and how they impact on the Au uptake kinetics?
4. Why ascorbic acid reduced GO outperformed other reductants? This need to be explained in a clear manner.

Reviewer #2 (Remarks to the Author):

This manuscript reports a method of gold recovery from $[\text{AuCl}_4]^-$ solution by reduced graphene oxide via reduction. Overall, the research is systematic and lots of experiments had been done by the authors. However, this manuscript is not appropriate for Nature Communication, due to the insufficient of innovation and the blurry reduction mechanism of Au ions. Detailed comments are listed as follow:

1. The strategy of gold recovery from thiosulfate solution by reduction reaction have been proposed by Jia et al. in 2020 (ACS Sustain. Chem. Eng. 2020, 8, 3673-3680; Chem. Eng. J. 2020, 394, 124866.), and the efficient and selective recovery of gold from $[\text{AuCl}_4]^-$ solution by MoS₂ has been reported as well in 2018 (Chem. Eng. J. 2018, 350, 692-702).
2. The reduction mechanism of $[\text{AuCl}_4]^-$ by rGO may be a little blurry, due to lack of more direct and powerful proofs for the origin and transfer path of electrons for Au ions reduction.
3. Why the gold extraction capacity of rGO synthesized by authors is much higher than the pristine GO and commercial graphene, and the difference between rGO and commercial graphene should be explained.
4. The e-waste was leached by authors with aqua regia, whereas the common gold leaching agent is cyanide and thiosulfate, thus the practical application potential of the proposed gold extraction process should be evaluated.
5. The structure and chemical property of different gold complex ion is quite different, which seriously affect the gold extraction capacity. Thus, the caparison of the extraction capacities of various gold adsorbents may be not objective enough (Table S1), due to the neglect of the gold complex ion form in different references.

Reviewer #3 (Remarks to the Author):

Dear Author,

Find the comments, suggestions and doubts raised. Unfortunately, it is not possible to support the publication of your work in the current form.

Major

1. The pH experiment is not properly described. Fig 1b (inset) shows a constant extraction

efficiency, however, Fig. S2 shows a changing on the extraction efficiency as a function of pH. Then, what is the difference between them? I am not convinced that there is no effect of acidic and basic pH in Fig. 1b inset due to Coulomb attraction and repulsion interactions with rGO charged surface, as increasing pH.

2. 1 g of rGO seems to be a very excessive amount for the extraction of gold or any other element. Although the cost of rGO decreases over the years, its real application is limited to the laboratory. So how to avoid this fact?

3. The main results of the work are based on 60 degrees, but this call for an extra process, in a real application how to achieve this efficiently? In any case, the results also must be highlighted at room temperature.

4. The broad band from 3800 to 3200 cm^{-1} is not attenuated after the reduction process using ascorbic acid (Fig. S3b). Why? Really no effect on hydroxyl groups (or water molecules)? Or how was the characterization done?

5. Can the authors describe how the dispersibility of rGO provides the abundant graphene-like areas? What is the connection between them? Testing the interaction of Au ions with rGO and pristine graphene does not support this idea. Instead, it conflicts with what is claimed.

6. Although the reliability of the results is not criticized, a high selectivity of as-made rGO for Au seems doubtful, considering other cationic and anionic elements. This fact is not discussed in detail and lacks an adequate explanation.

7. In conventional DFT computations a neutral system is needed. How is the charge distribution addressed in periodic computations, taking Au^{3+} with the proposed model? I recommend to compare the obtained results with gas-phase computations, for instance, as implemented in Gaussian.

8. The work lacks a state of the art on the application of rGO or other adsorbents to extract Au at the beginning of the work. A discussion section is also needed to guide non-proficient readers properly since the work presents results from various graphene-like materials.

9. The focus on e-water is not appropriate, since e-water extracted from the contaminated site is not used, instead it emulates. It is recommended to mention as a potential application.

10. Will it be possible to complete the kinetic analysis with the intraparticle diffusion study? It appears that the extracted Au remains on the rGO surface regardless of time, which will have a negative impact on the rGO saturation.

Minor

1. In Fig. S1a the reported and published data cannot be properly appreciated (green region). Improve the result presentation.

2. Pp. 3, Line 59: 10 ppt?

3. The gold extraction value seems to be different from that is reported at 10 min. Double check

4. Try to improve the resolution of images as much as possible.

Reviewer #4 (Remarks to the Author):

This paper reports a highly impact extraction behavior of Au ions with reduced graphenes. Then, this paper should be publishable on Nature Communications.

However, I have several questions before the publication recommendation.

1. Although the experimental results are really astonishing, the reviewer cannot understand sufficiently the extraction mechanism. The absolute extraction or adsorption amount of metallic Au is really huge, corresponding to multi-layer amount on the graphene surfaces and/or highly dense filling in the pores between the layer structures. The successive and efficient reduction mechanism of Au ions even on the Au-coated graphene surfaces or pore walls may be discussed.

2. I expect the explanation of the observed extraction ability of reduced graphenes from the structural aspects. Is there any locally ordered stacking structure for Au-"intercalated or doped" graphenes ?

3. You determined the activation energy using Arrhenius equation for the relationship between the extraction amount and measuring temperature.

Probably authors determined the equilibrium extraction amount. In such a case, the obtained data

must be treated thermodynamically, not kinetically. van't Hoff equation must be applied to a thermodynamic process to obtain the enthalpy difference. Authors could check this. The enthalpy change is 0.12 eV (about 10 kJ/mol), being not absolutely large, but not so small compared with the thermal energy.

The exponential factor of 0.12 eV is not necessarily large. However, authors obtain highly selective Au ion extraction. Then, the mechanism may be quite new. I hope that authors could consider this factor.

4. Au induces SERS and authors can detect sensitively the structural information on graphenes around Au nanoparticles. This could give valuable information on the mechanism.

5. Reduced graphenes have no high electron mobility. However, your results indicate highly efficient reduction of Au ions with electron transfer.

This may be interpreted.

6. How about the pore structural change during conversion of Au ions to metallic Au on graphenes? Ordinary reduced graphenes have almost no porosity from N₂ adsorption at 77 K. However, this sample should have unique porosity due to inclusion of metallic Au particle at the initial stage. Do you have the data?

Reply to reviewers' comments

We would like to thank all reviewers for careful reading of our manuscript and their reports. We have found them fair and useful. All the suggestions are incorporated in the revised manuscript as specified below. We have highlighted the revised parts in main text and SI in blue, so the corresponding changes are traceable.

To aid the editor and reviewer to clearly track our response, we have highlighted the reviews' comment in italic and blue, our response in black.

Reviewer #1 :

In the present manuscript, the authors reported and demonstrated that rGO enables effective and selective Au uptake from various water matrices. Various advanced characterization techniques were employed to verify the effectiveness of rGO for the reductive adsorption of Au. Although some interesting results were present, I still believed that this study fail to provide enough new insights on the underlying working mechanism. The reported sorption capacity and selectivity were also comparable or only slightly better than few recent publications (both batch and flow-through operations). I, hence, cannot recommend its publication in Nature Communications.

Reply: We thank the reviewer's positive comment on our work. The reviewer has raised two major issues in this manuscript, one is on the underlying mechanism, the other is the performance of rGO "the gold extraction capacity and selectivity are only comparable or slightly higher than existing research."

For the mechanism, we have carried out extra experiments to support our explanation and rewritten the mechanism study part (Page 6-10). A summarized mechanism can be found in the last paragraph of page 10. To aid the understanding of the mentioned structure, a schematic drawing of rGO was added (Fig. 2h). We believe it is clear in the modified version.

The mechanism can be briefly described as follow:

1. **The adsorption is driven by the concentration difference (ΔC) of adsorbates in its solution and on the adsorbent** (Science 374, 1215-1221 (2021).). In our case, $\Delta C = C_{sol}^{Au\ ion} - C_{rGO}^{Au\ ion}$, Once gold ions were adsorbed on rGO, >95% of them were reduced to Au⁰ spontaneously. Such rapid and complete conversion of gold ion to Au⁰, leaves an extremely low $C_{rGO}^{Au\ ion}$, consequently, **a high ΔC was maintained during the adsorption process. Therefore, rGO overcame the fast equilibrium at low C, and showed an ultrahigh adsorption capacity at even ppb level.**

2. To study why rGO can highly efficiently reduce gold ions? We considered the major atomic structures of rGO, one is the graphene area and the other is oxidized regions (See the schematic drawing Fig. R1).

Fig.R1 Schematic drawing of the atomic structure of rGO. The drawing is an artistic analogue of the rGO structure observed by high-resolution transmission electron microscope in previous report (Adv. Mater. 2010, 22, 4467–4472).

The reductive adsorption was enabled by the graphene area of rGO, it donated electrons to the adsorbed gold ion and reduced it to Au⁰. In addition, the wrinkles and warped area of graphene prompted the adsorption of gold ions and electron transfer to gold ions during the redox reaction, further increasing the gold extraction capacity.

This part of the mechanism has been supported mainly by

- (1) previous reports on gold doped graphene grown by chemical vapor deposition, in which a high concentration of gold ion was found to be reduced by graphene in a very short time (one to few minutes), which is explained by a redox reaction between graphene and gold ions.
- (2) Gold extraction by mechanically exfoliated graphene and simulation.
- (3) Electron transfer identified by spectroscopic analysis of rGO before and after gold extraction.

3. The oxidized regions of rGO, provided a good dispersibility of rGO, therefore allowing efficient adsorption and reduction of gold ions by graphene area, leading to an ultrahigh extraction capacity.

This part of the mechanism has been supported by:

- (1) Commercial graphene powder showed a low extraction capacity, suggesting graphene area is not the only factor for high extraction capacity, and a good dispersibility is also essential.
- (2) Control of oxidized region of rGO. Excessive elimination of the oxidized region led to agglomeration of rGO, decreased the dispersibility, and disabled some of the recovered graphene areas from being exposed to the aqueous gold ions, resulting in ineffective gold extraction.
- (3) Similar gold extraction capacities were observed for rGO reduced by hydrazine and hydroquinone. Both rGO materials have graphene areas recovered by reduction and show good dispersibility.

For the comment on the extraction capacity and selectivity, we kindly disagree. We have addressed as follows,

1. Our extraction capacity was compared with existing publications (from 2006-2021), and we note the comparison of extraction capacity is only fair when they are obtained at same or similar adsorbate concentration, as extraction capacity decreases when adsorbate concentration decreases. As the reviewer has pointed out, similar extraction capacity has been reported previously, but they all have measured at 100 ppm or higher level (Fig. S1 and table S1). In comparison, our reported extraction capacity is obtained at one order of magnitude lower concentration, specifically at 10 ppm, 1ppm even 100 ppb level. In SI Fig. S1 and table S1, we have listed all the reported capacity versus the concentration. At the same concentration of 10 ppm and lower, which is relevant to the concentration in e-waste leachate, our gold extraction capacities are 1850 mg/g and 1180 mg/g at 10 ppm and 1 ppm, respectively, this is at least 10 times higher than the previously reported results.
2. Regarding to the selectivity, we examined gold extraction selectivity both in simulated mixture of different metal salts (Fig. S13) and in the practical e-waste leachate, we have achieved 99% gold extraction without detectable adsorption of co-existing metals (adsorption in e-waste leachate then soaked in pH=1 HCl, page 13 in main text and Fig. S15). In contrast, previous studies reported preferential adsorption of gold, but their adsorption to other metals is unneglectable. The only gold adsorption without detectable adsorption to co-existing ions from a real e-waste leachate is, as far as we are aware, not reported elsewhere.
3. We have added a discussion on “state-of-art gold adsorbents” in “introduction” section main text, in which, abovementioned challenges have been summarized so it is easier to compare them with rGO.

My detailed comments are given below:

1. Similar reductive adsorption mechanism toward Au uptake were reported. The new insights need to be clearly illustrated and the novelty need to be justified as well.

Reply: Thanks for the reviewer’s comments and suggestions on this study. As reviewer has commented and also we have mentioned in the previous version, reductive adsorption is not new, indeed, this has been found in many novel gold adsorbents.

In the revised version, we have further emphasized this point in the introduction section when discussing the state-of-art gold adsorbents (Found in the first paragraph of the “introduction” section). Here, we would like to address the reviewer’s concern on the new insights and novelty in two folds.

1. New insights on the mechanism gained in this report.
 - (1) We have identified a clear-cut fact that, >95% gold adsorption of rGO is reductive adsorption, this quantitative result is new and exciting. Previously reported adsorbents showed a mixed form of adsorption despite a reductive adsorption is involved. This allowed the ultrahigh extraction capacity to trace amount of gold as it maintained a low adsorbate (gold ion) concentration on the rGO, which has discussed in details at the start of our reply to reviewer.

(2) The predominant reduction of Au^{3+} to Au^0 happens spontaneously, and supported the observed ultrahigh gold extraction capacity to trace amount of gold. In addition, this allowed us to reveal a site-specific adsorption for gold and other co-existing ions, that was, adsorption of gold ions mainly happened on the graphitic area of rGO, but adsorption of co-existing ions happened on the oxidized region of rGO by electrical charge based interaction. Such site-specific adsorption allows the precise selectivity of rGO, that was, when rGO was protonated, the functional groups lost their charge. Therefore, the previously adsorbed co-existing ions were stripped but not affect the gold adsorption.

(3) Another insight is on, how to achieve an ultrahigh gold extraction capacity?

For the selection of gold ions, gold ions with reductive potential >0.8 eV can be reduced and extraction. For the selection of graphene adsorbent, not all graphene-based materials can efficiently extract gold (Fig 2). The graphene area of rGO reduces gold ions, and the oxidized region of rGO provides dispersibility that the graphene area can efficiently reduce gold ions. Both are important for a high extraction capacity.

2. Novelty related to the mechanism

(1) The 95% reductive adsorption means nearly all gold ions are adsorbed and reduced to Au^0 by rGO. Though previous reports reported reductive adsorption, there is a significant amount of ionic gold adsorbed by the adsorbent. When recycling the mixed form of gold adsorbed on the adsorbent into the final elemental gold, the ionic gold needs to be chemically reduced to reach full extraction capacity. This requires further energy and resource. This is not needed as demonstrated in our manuscript. We believe this adds the novelty for the developed process (Main text page 6, end of the first paragraph).

(2) **This work proposes that graphene based materials**, especially, commercially, mass-produced, low-cost GO, could be used for practical application for **recycling gold from e-waste with ultrahigh capacity, precise selectivity, and economic viability**, we believe this is another novelty regarding to the potential impact of this work, and should be broadly interested by both the related research community and the related industry.

(3) We have added a paragraph discussing state-of-art gold adsorbents (paragraph 1 and 2 in “introduction” section, page 3), we believe this comparison with existing reports would manifest our novelty.

2. The Au sorption isotherm and kinetics need to be carried out systematically and fairly compared with previous reports (under normalized conditions).

Reply: Thanks for the reviewer's comments. We have plotted the isotherm and added a paragraph to describe the kinetics at different equilibrium concentration and at different extraction time, which has been normalized with full capacity (This has been added as Fig. S3 and the corresponding statement could be found in SI page 5).

The gold extraction capacities of the previous reports are plotted versus concentration (Fig. S1), and the concentration, type of gold ions, and corresponding extraction capacity were listed in Table S1, we believe this has made a fair comparison, and well supports claimed ultrahigh gold extraction capacity and rapid adsorption kinetics of rGO.

3. The authors claimed that “the areal gold extraction efficiency also depend on other factors”, what are these other factors and how they impact on the Au uptake kinetics?

Reply: Thanks for the reviewer's comments on this. We apologize for not being clear on this statement.

We have rewritten this part (first paragraph, page 8 of the main text). Briefly, we found the wrinkles and warped area, which are strained, showed more gold extraction. This was then explained using our theoretical calculation that the strained area both decreased the adsorption energy barrier and prompted electron transfer needed for redox reaction (SI supplementary section 4).

4. Why ascorbic acid reduced GO outperformed other reductants? This need to be explained in a clear manner.

Reply: Thanks for the reviewer's comments and suggestions. We apologize for the misunderstanding.

Firstly, for different reductants, in the original version of the manuscript, we have showed that, hydrazine (~1,835 mg/g from C = 10 ppm) is as good as ascorbic acid, the hydroquinone reduced GO showed slightly lower extraction capacity, which is explained by its relatively poor dispersibility. This has been discussed in Supplementary section 5 page 15.

Secondly, we also compared rGO obtained by various reduction time using ascorbic acid. This was added in the revised manuscript. Again, such difference is related to the different roles of graphene area and oxidized regions of rGO, and this has been explained at start of this reply, and details are discussed in page 9 of main text. We believe this now has been clearly explained.

Reviewer #2 :

This manuscript reports a method of gold recovery from [AuCl₄]⁻ solution by reduced graphene oxide via reduction. Overall, the research is systematic and lots of experiments had been done by the authors. However, this manuscript is not appropriate for Nature Communication, due to the insufficient of innovation and the blurry reduction mechanism of Au ions. Detailed comments are listed as follow:

Reply: We thank the reviewer's comment. To clarify "insufficient of innovation", we would like to use the summarized novelty and insights of this paper to address this concern.

1. New insights on the mechanism gained in this report.

(1) We have identified a clear-cut fact that, >95% gold adsorption of rGO is reductive adsorption, this quantitative result is new and exciting. Previously reported adsorbents showed a mixed form of adsorption despite a reductive adsorption is involved. This allowed the ultrahigh extraction capacity to trace amount of gold as it maintained a low adsorbate (gold ion) concentration on the rGO, which has discussed in detail at the start of our reply to reviewer.

(2) The predominant reduction of Au³⁺ to Au⁰ happens spontaneously, and supported the observed ultrahigh gold extraction capacity to trace amount of gold. In addition, this allowed us to reveal a site-specific adsorption for gold and other co-existing ions, that is, adsorption of gold ions mainly happened on the graphitic area of rGO, but adsorption of co-existing ions happened on the oxidized region of rGO by electrical charge based interaction. Such site-specific adsorption allows the precise selectivity of rGO, that is, when rGO is protonated, the functional groups lost their charge. Therefore, the previously adsorbed co-existing ions were stripped but not affect the gold adsorption.

(3) Another insight is on, how to achieve an ultrahigh gold extraction capacity?

For the selection of gold ions, gold ions with reductive potential >0.8 eV can be reduced and extraction. For the selection of graphene adsorbent, not all graphene-based materials can efficiently extract gold (Fig 2). The graphene area of rGO reduces gold ions, and the oxidized region of rGO provides dispersibility that the graphene area can efficiently reduce gold ions. Both are important for a high extraction capacity.

2. Novelty related to the mechanism

(1) The 95% reductive adsorption means nearly all gold ions are adsorbed and reduced to Au⁰ by rGO. Though previous reports reported reductive adsorption, there is a significant amount of ionic gold adsorbed by the adsorbent. When recycling the mixed form of gold adsorbed on the adsorbent into the final elemental gold, the ionic gold needs to be chemically reduced to reach full extraction capacity. This requires further energy and resource. This is not needed as demonstrated in our manuscript. We believe this adds the novelty for the developed process (Main text page 6, end of the first

paragraph).

(2) **Graphene based materials**, especially, commercially, mass-produced, low-cost GO, could be used for practical applications for **recycling gold from e-waste with ultrahigh capacity, precise selectivity, and economic viability**. We believe this is another novelty regarding to the potential impact of this work, and should be broadly interested by both the related research community and the related industry.

(3) We have added two paragraphs discussing state-of-art gold adsorbents (paragraph 1 and 2 in “introduction” section, page 3). We believe such comparison with existing reports would manifest the novelty of this paper.

As the mechanism has been commented in the following comments, we have addressed this in the reviewer’s comment No.2.

1. The strategy of gold recovery from thiosulfate solution by reduction reaction have been proposed by Jia et al. in 2020 (ACS Sustain. Chem. Eng. 2020, 8, 3673-3680; Chem. Eng. J. 2020, 394, 124866.), and the efficient and selective recovery of gold from [AuCl₄]- solution by MoS₂ has been reported as well in 2018 (Chem. Eng. J. 2018, 350, 692-702).

Reply: Thanks for the reviewer’s comments. The references that the reviewer listed are indeed very important. We have already cited one of these papers in our previous version, and the rest of the listed references are cited in the modified version as well. The development and challenges of these and other previously reported gold adsorbents are summarized as state-of-art gold adsorbents in the introduction part (paragraph 1 and 2 in Page 3), and their performances were now compared in Fig. S1 and Table S1. To more specifically address this comment, as shown in Table R1, we have compared mentioned adsorbents with our results. It is clear that, at a same/similar concentration range, rGO showed significant higher gold extraction capacity. Regarding to selectivity, though selective recovery of gold was reported, the only gold adsorption without detectable adsorption to co-existing ions from real e-waste leachate is, as far as we are aware, not reported elsewhere.

Finally, we note it is a misunderstanding to summarize the novelties of this work as proposing a reductive adsorption mechanism. We have summarized the novelties of this paper in detail at the start of our reply to the reviewer’s comment, and a similar discussion was added in the revised version in the “introduction” section of main text.

Table R1 The comparison of mentioned adsorbents with our results

Gold adsorbent	Gold ion concentration (ppm)	Extraction capacity (mg/g)	Adsorption mechanism	Remarks
MoS ₂ /chitosan	156	1100 (maximum result)	Photochemical reduction	
	10	<50		
MoS ₂ /ZnS	100	1120 (maximum result)	Photochemical reduction	
	50 (lowest concentration tested)	500		
MoS ₂	450	1133 (maximum result)	S-Au interaction	elution and further reduction gold is needed
Our work rGO	10	1850	Spontaneous redox reaction	
	1	1180		
	0.1	690		

2. The reduction mechanism of [AuCl₄]⁻ by rGO may be a little blurry, due to lack of more direct and powerful proofs for the origin and transfer path of electrons for Au ions reduction.

Reply: Thanks for the reviewer's comment. We have re-written the mechanism part in page 5-10 in the revised manuscript (main text). Briefly, regarding the proof of origin and transfer path of electrons, (1) we have identified graphene area of GO is responsible for electron transfer. The graphene area (origin of electron) transfer electron to gold ion and reduce Au³⁺ to Au⁰. Such redox reaction and the electron transfer process have been widely studied in previous research on using gold to tune the work function of chemical vapor deposited graphene. It has been clearly pointed out that, the redox reaction between gold ion and the graphene, and the electron transfer from graphene to gold ion are responsible for reduction of gold ion by graphene (ACS Nano 4, 2689-2694 (2010); ACS Nano 4, 4595-4600 (2010)). In our study, this has been proved by the experiments of mechanically exfoliated graphene, and we believe this is the direct evidence that graphene is responsible for the gold reduction. (2) Such redox reaction is further testified by gold ions with different reduction potentials. (3) Furthermore, we have probed the electron transfer from rGO to gold by Raman and UV-Vis spectroscopy. The related discussion and evidence could be found in the last paragraph of page 8 main text, and in supplementary information page 7-9.

3. Why the gold extraction capacity of rGO synthesized by authors is much higher than the pristine GO and commercial graphene, and the difference between rGO and commercial graphene should be explained.

Reply: Thanks for the reviewer's comments. We found that addressing this comment is very important to help our potential readers understand the uniqueness of rGO for gold extraction.

Two major differences between commercial graphene and rGO are identified, (1) the former has more graphene areas which are beneficial for gold reduction. This is supported by our Raman analysis. But, (2) commercial graphene shows poor dispersibility due to a lack of functional groups, as seen from the picture that the commercial graphene either floated on the gold solution or settled at the bottom of the gold solution (Supplementary information Fig. S12). A good dispersibility allows abundant graphene area, the redox site, to become sufficiently exposed/accessible to aqueous gold ions to initiate reduction. Thus poor dispersibility of commercial graphene because of a lack of functional groups leads to a poor extraction capacity. Regarding to GO, it is known to have limited amount of graphene areas, and was proved by our newly added XPS analysis, therefore, GO could not efficiently reduce gold ion, leading to a little reductive adsorption capacity.

The above discussion has been included in the mechanism study section in the main text on page 9-10, and Fig. 2h has been added as a visual comparison of structural differences among GO, rGO and commercial graphene.

4. The e-waste was leached by authors with aqua regia, whereas the common gold leaching agent is cyanide and thiosulfate, thus the practical application potential of the proposed gold extraction process should be evaluated.

Reply: Thanks for the reviewer's comments. The cyanide and thiosulfate are two popular leaching agents. However, the aqua regia based leaching method is evaluated as a flexible, low-cost, and sustainable method environment-friendly leaching method (ACS Sustainable Chem. Eng. 2021, 9, 2129–2135; Waste Manag. 45, 258-271 (2015); Hydrometallurgy 115-116 (2012) 30–51; Journal of Hazardous Materials 164 (2009) 1152–1158; Proc. Natl. Acad. Sci. U.S.A. 117, 16174-16180 (2020); J. Am. Chem. Soc. 140, 16697-16703 (2018)). Therefore, it is a widely adopted process in the industry. This is further supported by the fact that most publications on gold recycling from e-waste used aqua regia as leaching agent, this can be seen from the long reference list in this manuscript.

In our study, defined by the reduction potential, rGO is incapable of recycling gold cyanide and gold thiosulfate. We have added a statement as evaluation at the end of the

second paragraph on page 7: “ For ions with $E_0 < 0.8$ eV, for example, $[\text{Au}(\text{CN})_2]^-$ and $[\text{Au}(\text{S}_2\text{O}_3)_2]^{3-}$, rGO showed little extraction, corroborating our conclusion that the redox reaction between gold ion and rGO dictated the reported behaviour. ”

5. The structure and chemical property of different gold complex ion is quite different, which seriously affect the gold extraction capacity. Thus, the comparison of the extraction capacities of various gold adsorbents may be not objective enough (Table S1), due to the neglect of the gold complex ion form in different references.

Reply: Thanks for the reviewer’s comment and suggestion. We have added the information of the gold complex ion in our comparison in Table S1. As can be seen, in Table S1, only one cited research used $\text{Au}(\text{S}_2\text{O}_3)_2^{3-}$, the rest of the publications have used $[\text{AuCl}_4]^-$ or Au^{3+} , which rGO could efficiently reduce and extract. This echoes our reply to the reviewer’s comment No. 4.

Reviewer #3 :

Find the comments, suggestions and doubts raised. Unfortunately, it is not possible to support the publication of your work in the current form.

Major:

1. The pH experiment is not properly described. Fig 1b (inset) shows a constant extraction efficiency, however, Fig. S2 shows a changing on the extraction efficiency as a function of pH. Then, what is the difference between them? I am not convinced that there is no effect of acidic and basic pH in Fig. 1b inset due to Coulomb attraction and repulsion interactions with rGO charged surface, as increasing pH.

Reply: Thanks for the reviewer's comments and suggestions. We apologize for the misunderstanding raised here. For Fig. 1b, we used excessive rGO to reach high extraction efficiency (weight ratio rGO: Au=50:1), this is a commonly-used protocol for determination of efficiency (J. Am. Chem. Soc. 140, 16697-16703 (2018); Proc. Natl. Acad. Sci. U.S.A. 117, 16174-16180 (2020); Chem. Eng. J. 255 (2014) 97–106; Carbohydrate Polymers 286 (2022) 119307). For the extraction capacity, we have kept the weight ratio rGO: Au=1:2. Using excessive adsorbates to measure the extraction capacity has been widely used in previous reports (ACS Sustainable Chem. Eng. 2020, 8, 3673–3680); Talanta 93 (2012) 350–357; Chem. Eng. J. 255 (2014) 97–106; ACS Sustainable Chem. Eng. 2021, 9, 2129–2135). All these information is now available in “Method” section in the main text.

Regarding to pH, as the reviewer has pointed out and we have stated in the previous version, the measured extraction capacity is affected by pH. We have changed the writing to make this clearer to the reviewer. The corresponding statement can be found in paragraph 2 in the main text on page 5.

2. 1 g of rGO seems to be a very excessive amount for the extraction of gold or any other element. Although the cost of rGO decreases over the years, its real application is limited to the laboratory. So how to avoid this fact?

Reply: Thanks for the reviewer's comment.

1. Commercial GO was directly used in this study without further processing, and this statement could be found at the start of first paragraph in “Highly efficient gold extraction by rGO” section main text and in “methods” section. As far as we know, our GO provider uses GO as starting materials for industrial applications, for example, thick graphene film (>100 um) for thermal management needed in 5G mobile phones. In 2013, the production capacity of GO in one Chinese company reached 100 tonnes per year (Nat. Nanotech., 2014, 9(10), 726). Therefore, in our case, 1g rGO would be excessive regarding its availability.
2. Regarding to the cost, as mentioned in the main text, the cost of GO is less than 0.5 RMB/g. In comparison, the price of gold is >300 RMB/g. With the reported

extraction capacity, the cost of GO or rGO should not be a concern. We have discussed this in the last paragraph on page 14 of the revised version.

3. Regarding the real application of graphene is limited to the laboratory, we believe this is a topic that better discussed elsewhere. Nevertheless, as far as we know, the development of graphene has extended its territory globally from lab to industrial scale manufacturing and applications. This can be found in these two papers published in National Science Review and Nature Nanotechnology (Natl. Sci. Rev., 2018, 5(1),90, and (Nat. Nanotech., 2014, 9(10),726).
4. Specifically, in this study, we have demonstrated scalable preparation of rGO suspension and fabrication of rGO membranes (Fig. 4a), and are applying two patents based on this work. The scale-up of manufacturing of reported gold adsorbents should be feasible, and this is because GO and ascorbic acid are readily commercially available. The process parameters are mild (reduction of GO operated at 95 °C and ascorbic acid is a green chemical). We are now setting up a pilot-scale trial and working with the local e-waste recycling industry for applications. We believe the above facts, consideration, and continuing efforts taking currently can address the reviewer's concern "how to avoid this".

3. The main results of the work are based on 60 degrees, but this call for an extra process, in a real application how to achieve this efficiently? In any case, the results also must be highlighted at room temperature.

Reply: We thank the reviewer's comments. We believe there is a misunderstanding, the main results of this work are based on 25 degrees. The result based on 60 degrees is only mentioned in the introduction part and in the section that we studied the temperature-dependent extraction capacity.

In the previous version, we have noted the temperature information in the "method" session, reviewer's comment makes us aware that this might mislead our potential readers as well. Therefore, in the modified version, we have added temperature information "all the experiments were performed at 25 °C" at the end of each figure caption in the main text.

4. The broad band from 3800 to 3200 cm⁻¹ is not attenuated after the reduction process using ascorbic acid (Fig. S3b). Why? Really no effect on hydroxyl groups (or water molecules)? Or how was the characterization done?

Reply: Thanks for the reviewer's suggestion. We apologize for this mistake in presenting the data.

In the previous version of FTIR curves, the scale of each curve was plotted at various intensity scales, which makes seemingly not attenuated curve of rGO. We have re-

plotted all three curves at the same intensity scale. The broad band from 3800 to 3200 cm^{-1} has been clearly attenuated after reduction (Fig. R2b). The detail has been modified in the revised manuscript.

Fig. R2| Characterization of GO, rGO, and rGO-Au. (a) Their XRD patterns and (b) FTIR analyses. (c) TG and differential scanning calorimetry (DSC) curves of KAuCl₄, rGO, and (d) rGO-Au after 24 h extraction (rGO-Au-24 h) measured in air.

5. Can the authors describe how the dispersibility of rGO provides the abundant graphene-like areas? What is the connection between them? Testing the interaction of Au ions with rGO and pristine graphene does not support this idea. Instead, it conflicts with what is claimed.

Reply: We thank reviewer's comment. We have added a schematic drawing (Fig. 2h) to visualize the correlations among dispersibility, atomic structures and extraction capacities of rGO, pristine GO and commercial graphene. Specifically,

(1) Dispersibility does not provide abundant graphene-like area. Dispersibility is enabled by the oxidized region, while the reductive adsorption happens on graphene areas. Both dispersibility and the graphene areas are essential for the gold extraction. Briefly, more graphene areas should give higher adsorption capacity, as there are more redox sites providing electrons for reduction. However, these graphene redox sites are not accessible/exposed to aqueous gold ions without dispersibility. This leads to a balance between dispersibility and graphene area on the resulting extraction capacity. This is validated by controlling the reduction of (r)GO which essentially, tuned the graphene area and the oxidized area. In the modified version, we have rewritten this part to explain this (page 9 in the main text) and used Fig. 2h to visualize the relations between graphene areas, oxidized region and dispersibility.

(2) Experiment performed with mechanically exfoliated graphene does not conflict with rGO. Instead, support each other. Briefly, the main purpose of the experiments on

exfoliated graphene is to prove that it is graphene area that dominates the redox reaction and reduces gold ions. We note that the high extraction capacity observed on the mechanically exfoliated graphene, is obtained by immersing several mono- and few-layered graphene flakes on Si substrate in the aqueous gold solution (see SI section 4). If only considering the contribution of graphene area in observed ultrahigh gold extraction capacity, the mechanically exfoliated graphene would be the best choice for gold extraction as all of its graphene area can be used for gold extraction. However, we have tried commercial graphene, which is mass produced, a critical feature for practical gold extraction as large amounts of adsorbents are needed, and, compared to rGO, has similar structure to mechanically exfoliated graphene, but found low extraction capacity. This allowed us to further reveal that, both the graphene area and the oxidized region are essential for practical gold extraction. The graphene area reduces gold ions, but the oxidized region provides dispersibility that the abundant graphene area of rGO, is sufficiently exposed to aqueous gold ions, reaching a high extraction capacity. We have rewritten this part on page 9-10, so the critical roles of both the graphene area and oxidized regions in the gold extraction now are better described.

6. Although the reliability of the results is not criticized, a high selectivity of as-made rGO for Au seems doubtful, considering other cationic and anionic elements. This fact is not discussed in detail and lacks an adequate explanation.

Reply: We thank the reviewer's comment. The high selectivity has been measured using the same methods applied for measuring gold extraction capacity. We note that the as-made rGO showed selectivity, but such selectivity can be further improved by (de)protonation method.

The high selectivity is realized based on the site-specific adsorption for gold (on graphene area) and other cationic elements (on functional groups of the oxidized region). (1) As gold is reduced by the graphene area of rGO, other metal cations are reported to be adsorbed on the oxygen-containing group in the oxidized region mainly by electrostatic interaction. We have exploited this difference, for example, after rGO's adsorption to gold ions and its co-existing metal cation, and added HCl to protonate the oxygen-containing functional groups. This stripped off the co-existing metal cations that attached to the functional groups without affecting gold adsorption as it has been reduced to Au⁰ on the graphene area, therefore, achieving a high selectivity. This conclusion has been drawn based on a comparison of before and after the protonation process. We believe the high selectivity is solid. To better present this mechanism, we have visualized it in Fig. 3a, and a related discussion can be found in the section "Realization of a high selectivity for gold extraction" of the main text.

(2) Regarding to the anionic elements, we have examined Cl⁻, NO₃⁻, SO₄²⁻, [PtCl₄]²⁻,

these anions are presented in Cu, Ni, and Pt salts, and e-waste leachate. We did not find these anions influenced selectivity. This is probably because there is no specific interaction for these anions with either graphene or oxidized regions. We have added this information in Fig. S13 in Supplementary information.

(3) After burning gold loaded rGO, we obtained a high purity gold as justified by XRD and EDS. We believe this is the other strong evidence that there is little adsorption of either cation or anion. We have added this information on page 14, “Its EDS analysis confirmed >95% purity with the rest being carbon, oxygen and sodium (Fig. R3), this further proved that the adsorption of co-existing ions (both cationic and anionic ions) was little, in good agreement with the observed precise gold selectivity of rGO.”

All discussion on the selectivity performance and related explanation is now in the section “Realization of a high selectivity for gold extraction” of the main text.

Fig. R3| Extraction selectivity of rGO. (a) Extraction efficiency for an equal-part mixture of [AuCl₄]⁻, Cu²⁺, Ni²⁺ and [PtCl₄]²⁻ ions. Inset: Same for aqueous solutions containing individual metal ions. (b) the influence of the co-existing cationic and anionic elements for gold extraction using rGO.

7. In conventional DFT computations a neutral system is needed. How is the charge distribution addressed in periodic computations, taking Au³⁺ with the proposed model? I recommend to compare the obtained results with gas-phase computations, for instance, as implemented in Gaussian.

Reply: Thanks for the reviewer’s comments and suggestions. As Gaussian is unavailable for us to perform the suggested computation (copyright limitation). Instead, we used VASP to perform gas-phase calculations on cluster models, from which the different charge states can be considered. Fig. R4 shows the structural models for Au-adsorbed graphene and carbon nanotube (CNT) clusters, which were saturated with H atoms. The +3 charge state of the cluster models was realized by artificially setting the number of electrons, and the compensating background charge was included to ensure

the convergence of electrostatic energy. After structural relaxation, the Bader charge analysis shows that the number of electrons transferred from CNT and graphene clusters to Au^{3+} was 3.03 and 2.81, respectively. Compared to graphene, CNT transferred 0.22 more electrons to Au^{3+} , which is consistent with the calculation based on AuCl_3 models (0.13 e).

Fig. R4 Top and side views of Au-adsorbed (a) graphene and (b) CNT clusters.

We have added above information on page 14 of the revised manuscript SI section 4.

8. The work lacks a state of the art on the application of rGO or other adsorbents to extract Au at the beginning of the work. A discussion section is also needed to guide non-proficient readers properly since the work presents results from various graphene-like materials.

Reply: Thanks for the reviewer’s comments and suggestions. This is very useful on explaining the novelty and importance of this work. The discussion of state-of-the-art gold adsorbents has been added in the “introduction” section in the revised manuscript.

The different graphene-like materials have been compared using Raman analysis in Fig. S15, and their structure-property correlation is discussed on page 9 in the main text.

9. The focus on e-water is not appropriate, since e-water extracted from the contaminated site is not used, instead it emulates. It is recommended to mention as a potential application.

Reply: We thank the reviewer’s comment. From the context, we believe the “e-water” that the reviewer suggested is the water discharged from the e-waste contaminated site. This is a misunderstanding, as “e-water” is not studied in our study, instead, e-waste leachate is used in our study.

Specifically, we have followed a typical e-waste recycling procedure used in many previously published papers (ACS Sustainable Chem. Eng. 2021, 9, 2129–2135; Proc. Natl. Acad. Sci. U.S.A. 117, 16174-16180 (2020). Chem. Eng. J. 255 (2014) 97–106),

that is, first leaching all the metals in the e-waste by aqua regia, then extracting gold ions in the leachate by rGO. With the reported high extraction capacity, selectivity, and potential economic benefits, we believe rGO can be used for application of e-waste recycling.

10. Will it be possible to complete the kinetic analysis with the intraparticle diffusion study? It appears that the extracted Au remains on the rGO surface regardless of time, which will have a negative impact on the rGO saturation.

Reply: We thank the reviewer's comment. Indeed, an additional kinetic study on the intraparticle diffusion study would be beneficial to gain a better understanding of rGO's gold extraction behavior. So we have tried to fit our data with the intraparticle diffusion model. We did not find linearity, suggesting rejection of the intraparticle diffusion model (Fig. R5).

Fig. R5 Fitting gold extraction capacity using an intra-particle diffusion model (Critical review in adsorption kinetic models. J. Zhejiang Univ. Sci. A 10, 716–724). t is the extraction time.

This is understandable, as intraparticle diffusion emphasizes diffusion of adsorbate from the surface to the inside of a pore, which is generally seen in porous adsorbents. However, this is not the case in our study. As a two dimensional material, rGO does not have intrinsic pores needed for intraparticle diffusion. Instead, gold ions were reduced on the surface of colloidal rGO nanosheets.

Regarding to the “extracted Au remains on the rGO surface regardless of time, which will have a negative impact on the rGO saturation.”. In a typical adsorption process, adsorbates, the stably adsorbed ones, will always be on the adsorbent regardless of time. This allows us to define the amount of adsorbate on the adsorbent, that is, adsorption capacity, when adsorption equilibrium is reached. In our case, we agree with the reviewer that gold remains on the rGO surface regardless of time (Fig. S7). But it is similar to other adsorbent or their adsorption process that adsorbates remain on the adsorbent surface regardless of time.

On the other hand, indeed, gold nanoparticles that remain on the rGO would cause

problems on the reuse of rGO. This has been discussed this on page 14 main text: “We also tried regeneration of rGO by dissolving gold extracted by rGO with thiourea/HCl, and found regenerated rGO showed a gold extraction capacity of ~1000 mg/g (T=25 °C). However, considering GO we used is a mass-produced commercial product which has a price of less than 0.5 RMB per gram (Shenzhen Matterene Technology), and the high temperature used for removing rGO is, nevertheless, used for gold melting purposes. Therefore, direct isolation at high temperature may be a better choice than regeneration with respect to the process cost and sustainability”.

Minor:

1. In Fig. S1a the reported and published data cannot be properly appreciated (green region). Improve the result presentation.

Reply: Thanks for the reviewer’s suggestions. We have listed all the data in Fig.S1a in Table S1, we have added a statement in Fig.S1 caption, “all the data compared in Fig.S1a was listed in Table S1”, so that these data are traceable to our reader. To improve this result presentation, we have added a plot comparing the reported data obtained at a concentration range from 10-20 ppm with our results. We believe this manifests the ultrahigh gold extraction performance of the rGO.

2. Pp. 3, Line 59: 10 ppt?

Reply: Thanks for the reviewer’s comments. ppt is for part per trillion. We have checked this carefully. The related results are discussed in the main text Page 4 and SI page 2.

3. The gold extraction value seems to be different from that is reported at 10 min. Double check

Reply: Thanks for the reviewer’s comments and suggestions. We believe this is a misunderstanding. We have checked this carefully.

Because the gold extraction reaches equilibrium only at 24 h, that is, at 10 min, the adsorption capacity is not yet reached its maximum extraction capacity. Therefore, it is sensible that “The gold extraction value measured at 24 h is higher than that is measured at 10 min.”

4. Try to improve the resolution of images as much as possible.

Reply: Thanks for the reviewer’s comments and suggestions. We have provided all four figures in high resolution at the end of the manuscript.

Reviewer #4 :

This paper reports a highly impact extraction behavior of Au ions with reduced graphenes. Then, this paper should be publishable on Nature Communications.

Reply: We thank the reviewer's positive comment

However, I have several questions before the publication recommendation.

1. Although the experimental results are really astonishing, the reviewer cannot understand sufficiently the extraction mechanism. The absolute extraction or adsorption amount of metallic Au is really huge, corresponding to multi-layer amount on the graphene surfaces and/or highly dense filling in the pores between the layer structures. The successive and efficient reduction mechanism of Au ions even on the Au-coated graphene surfaces or pore walls may be discussed.

Reply: We thank the reviewer's comment. We apologize for not being clear on the gold extraction mechanism. In our experiments of gold extraction by rGO colloid and mechanically exfoliated graphene, which are mono- or few-layered. We did not observe the gold filling in the interlayer/pore, and this is consistent with the previous report on gold doped graphene (ACS Nano 4, 2689-2694 (2010); ACS Nano 4, 4595-4600 (2010)). We have rewritten the mechanism part and believe it is clear now. The mechanism can be briefly described as follow:

1. The adsorption is driven by the concentration difference (ΔC) of adsorbates in its solution and on the adsorbent (Science 374, 1215-1221 (2021)). In our case, $\Delta C = C_{sol}^{Au\ ion} - C_{rGO}^{Au\ ion}$, Once gold ions were adsorbed on rGO, >95% of them were reduced to Au⁰ spontaneously. Such rapid and complete conversion of gold ion to Au⁰, leaves an extremely low $C_{rGO}^{Au\ ion}$. Consequently, **a high ΔC was maintained during the adsorption process. Therefore, rGO overcame the fast equilibrium at low C, and showed an ultrahigh adsorption capacity at even ppb level.**

2. To study why rGO can highly efficiently reduce gold ions? We considered the major atomic structures of rGO, one is the graphene area, and the other is oxidized regions (See schematic drawing Fig. R1).

The reductive adsorption was enabled by the graphene area of rGO, it donated electrons to the adsorbed gold ion and reduced it to Au⁰. In addition, the wrinkles and warped area of graphene prompted the adsorption of gold ions and electron transfer to gold ions during the redox reaction, further increasing the gold extraction capacity.

This part of the mechanism has been supported mainly by

(1) previous reports on gold doped graphene grown by chemical vapor deposition, in which high concentration of gold ion was found to be reduced by graphene in a very short time (one to few minutes), which is explained by a redox reaction between graphene and gold ions.

(2) Gold extraction by mechanically exfoliated graphene and simulation.

(3) Electron transfer identified by spectroscopic analysis of rGO before and after gold extraction.

3. The oxidized regions of rGO, provided a good dispersibility of rGO, therefore allowing efficient adsorption and reduction of gold ions by graphene area, leading to an ultrahigh extraction capacity.

This part of the mechanism has been supported by:

- (1) Commercial graphene powder showed a low extraction capacity, suggesting graphene area is not the only factor for high extraction capacity and a good dispersibility is also essential.
- (2) Control of oxidized region of rGO. Excessive elimination of the oxidized region led to agglomeration of rGO, decreased the dispersibility, and disabled some of the recovered graphene areas from being exposed to the aqueous gold ion, resulting in ineffective gold extraction.
- (3) Similar gold extraction capacities were observed for rGO reduced by hydrazine and hydroquinone. Both rGO materials have graphene areas recovered by reduction and show good dispersibility.

2. I expect the explanation of the observed extraction ability of reduced graphenes from the structural aspects. Is there any locally ordered stacking structure for Au-“intercalated or doped” graphenes?

Reply: We thank the reviewer's comment. We agree with the reviewer that the extraction ability stems from the structural aspects. As mentioned, the graphene area in rGO provides reductive adsorption. We have not observed intercalation of graphene as confirmed by XRD analysis (Fig. R6), as the intercalation will lead to the emergence of the diffraction peaks at a lower diffraction angle because of the expansion of interlayer spacing. This is also consistent with a previous report on gold doped graphene (ACS Nano, 20, 4, 4595). For the doping, as electrons are transferred from rGO to gold ions, we have found the blueshift of G peak from 1602 to 1606 cm^{-1} due to phonon stiffening, suggesting a p-doping and electron transfer from rGO to gold (Fig. S5). We have added the statement on “doping of rGO after gold extraction” in the last paragraph on page 8 of the main text and the last paragraph on page 7 of supplementary information, and Fig. S5 inset.

Fig. R6 XRD pattern of rGO after extraction with 1 ppm gold solution.

3. You determined the activation energy using Arrhenius equation for the relationship between the extraction amount and measuring temperature.

Probably authors determined the equilibrium extraction amount. In such a case, the obtained data must be treated thermodynamically, not kinetically. van't Hoff equation must be applied to a thermodynamic process to obtain the enthalpy difference. Authors could check this. The enthalpy change is 0.12 eV (about 10 kJ/mol), being not absolutely large, but not so small compared with the thermal energy.

The exponential factor of 0.12 eV is not necessarily large. However, authors obtain highly selective Au ion extraction. Then, the mechanism may be quite new. I hope that authors could consider this factor.

Reply: We thank the reviewer's comment. We followed the reviewer's suggestion and compared using the reviewer provided enthalpy with other gold extraction studies, which are from ~40-90 kJ/mol (J. Chem. Eng. Data 2013, 58, 209–216; J. Mater. Chem. A, 2020, 8, 3438–3449; Talanta 93 (2012) 350–357; Chem. Eng. J. 380 (2020) 122511). Indeed, the enthalpy is much lower than the existing report, suggesting that reductive adsorption is energy favored. This probably is due to (1) the 95% reduction to adsorbed gold ion by rGO, leading to a high concentration difference of gold ion in the solution and on the rGO surface, and (2) the wrinkles on the graphene area and large surface area of rGO are contributing to the high extraction capacity as well.

Regarding choosing the model of Van Hoff's or Arrhenius's, we considered that as extraction amount at equilibrium (extraction capacity) is a direct result of the rate of reduction reaction between rGO and gold ion, we believe this fits better with the Arrhenius's equation. Furthermore, consistent with the reviewer's suggestion, the activation energy calculated by Arrhenius's was then compared with typical chemisorption, which is 20.9–418.4 kJ/mol (J. Hazard. Mater. 143, 220-225 (2007)), suggesting a much lower activation barrier for the reduction reaction.

Followed the reviewer's comments, we have added a justification of choosing

Arrhenius's equation and the comparison of activation energy in the last paragraph page 6 of the main text.

4. Au induces SERS and authors can detect sensitively the structural information on graphenes around Au nanoparticles. This could give valuable information on the mechanism.

Reply: We thank the reviewer's comments. We have followed the reviewer's suggestion and compared the Raman fingerprints of rGO before and after gold extraction.

To gain statistical information on the structural change of rGO using Raman analysis, we have used Raman map to compare their Raman peaks before and after extraction. We found that the SERS is indeed evident, as the peak intensity of I_D and I_G both increased after gold extraction. Specifically, I_D peak shows an intensity range from 300-1400 for rGO, increased to 1500-2600. I_G peak increased from 300-1200 to 1400-2200, suggesting the significant gold assisted SERS effect. I_D/I_G decreased from a range of 1.00-1.15 to 0.93-1.02, such decrement in I_D/I_G , though is counterintuitive, suggested the increase of defects in the defective graphene materials (Nano Lett. 11, 3190-3196 (2011)), and again suggested the electron donation behavior of rGO. Furthermore, as stated in response to reviewer's comment No.2, we have observed the blueshift of G peak from 1602 to 1606 cm^{-1} , suggesting p-doping and the electron transfer from graphene to gold ions.

We note as the rGO is defected by its holes and oxidized regions, fine Raman fingerprint, for example, G' is not evident, and we did not find more structural information related to the change of G' peak. The three key pieces of information we have obtained from our Raman mapping are, (1) as suggested by the reviewer, there is indeed a significant SERS effect enabled by adsorbed gold. (2) The statistical result on I_D/I_G support our previous result on the increment of the defect in rGO after gold extraction, suggesting the electron transfer from rGO to gold ion. (3) The blueshift of G peak from 1602 to 1606 cm^{-1} due to phonon stiffening, suggesting a p-doping and electron transfer from rGO to gold.

All the updated Raman analysis was added in Fig. S5 and Fig. S6 in the supplementary information.

5. Reduced graphenes have no high electron mobility. However, your results indicate highly efficient reduction of Au ions with electron transfer. This may be interpreted.

Reply: We thank the reviewer's comments. Indeed, reduced graphene has relatively low electron mobility. Graphene areas on rGO are interconnected domains (Adv. Mater. 2010, 22, 4467-4472), giving electron mobility for rGO. The electron mobility

generally considers long-distance electron transport. In our case, once gold ion adsorbed on the graphene area of rGO, it was reduced back to Au⁰. From our SEM image (Fig. 2a and Fig. S7), each gold particle has an intra-particle distance between tens nanometers to a few hundred nanometers, this suggests the electron transfer needed for reductive adsorption may only require electron transfer in sub-micrometre range, so that the locally interconnected graphene regions are able to provide electron to gold ions adsorbed at its vicinity and achieve reductive adsorption.

We have added this statement in SI Page 9.

6. How about the pore structural change during conversion of Au ions to metallic Au on graphenes? Ordinary reduced graphenes have almost no porosity from N₂ adsorption at 77 K. However, this sample should have unique porosity due to inclusion of metallic Au particle at the initial stage. Do you have the data?

Reply: We thank the reviewer's comments. Followed the reviewer's suggestion, we measured the specific surface area of rGO before and after gold extraction, the pristine rGO showed a specific surface area ~12.5 m²/g, and after gold extraction, it decreased to ~1.5 m²/g (Fig. R7).

Fig. R7 The N₂ adsorption–desorption isotherms of (a) rGO and (b) rGO-Au.

The low specific surface area is partially in agreement with the reviewer's comment that there is almost no porosity of rGO, but also suggested little porosity in rGO after gold extraction. This could be understood by, because a lack of intrinsic pores of rGO, gold is adsorbed and reduced on the surface of single- or few-layered rGO, the surface decoration of gold on rGO should not generate porosity. Therefore, the little specific surface area is observed. The decrement in specific surface area of rGO after gold extraction could be qualitatively understood by that, the adsorbed gold, accounting for ~65 wt% of the measured sample (calculated based on an extraction capacity of ~1850 mg/g), but does not contribute much to the surface area, thus a decreased specific surface area was observed.

We note the specific surface area is measured in vacuum dried sample, in which, the stacking and densification of rGO nanosheets during the sample preparation are unavoidable, leading to a low specific surface area. Therefore, it does not reflect the available surface area of rGO in its colloid form for gold adsorption, in which rGO colloid is mostly mono- or few- layers, and its large surface area are readily available for gold extraction.

REVIEWER COMMENTS

Reviewer #1 (Remarks to the Author):

My previous comments were well-addressed by the authors in the new version. The authors have justified the novelty and clarified the underlying mechanism toward selective gold uptake. I, hence, glad to recommend its publication in NC in the present form.

Reviewer #3 (Remarks to the Author):

Dear Authors,

Thank you for considering the comments and suggestions provided. While considered point-by-point, these are not fully addressed from my point of view, some examples:

1. Concerning the amount of material, using a high amount of GO could be similar to using activated carbon. This is obviously easier and cheaper to produce. Although there is a large-scale production under development, the bulk-GO obtained could be different from the one reported in this study. Therefore, finding results similar to those reported here for large-scale water treatment is an ambitious bet.
2. The authors seem to be unclear about periodic calculations and molecular calculations (gas phase), the latter cannot be performed in VASP as discussed in the response letter. Although, in the revised manuscript, state-of-the-art density functional calculations are now correct, in the previous report, it was recommended to corroborate the theoretical predictions with molecular density calculations as a charged surface appears due to the presence of oxygen- functional groups (mainly hydroxyl groups) as well as the proposed Au system. For a charged system, the charge distribution is not correctly described by VASP or other related DFT software, see e.g., Physical Review Letters 126.7 (2021): 076401. Although this point is not mandatory, the authors should comment on the idealized and unrealistic system proposed by them.
3. For the study of intraparticle diffusion, see: Scientific Reports 12.1 (2022): 1-12. How to calculate and discuss. The idea is to evidence the 2 or 3 regions of linearity through a simple linear fit, showing the adsorption behavior (from kinetic data) on the surface or internal structure. More importantly, the authors can properly discuss the intraparticle diffusion rate (k_p) and the intercept (C) . The latter shows the boundary layer effect or surface adsorption. This point is not mandatory, but authors are encouraged to present this result for a better understanding of the adsorption process.

However, I recognize that the manuscript has improved remarkably, therefore I recommend its publication at the best criteria of the other Reviewers.

Reviewer #4 (Remarks to the Author):

Authors revised very nicely the manuscript and it has been highly improved. However, I did not know that similar phenomena were already published for MoS₂, when I reviewed before. I read the cited paper on MoS₂. Nature family journals should make much of the originality and creativity. The secondary study is less valued compared with the pioneering one. Therefore, I cannot recommend the publication.

Reply to reviewers' comments

We would like to thank all reviewers for careful reading of our manuscript and their reports. We have found them useful. All the suggestions are incorporated in the revised manuscript as specified below. We have highlighted the revised parts in main text and SI in blue, so the corresponding changes are traceable.

To aid the editor and reviewer to clearly track our response, we have highlighted the reviews' comment in italic and blue, our response in black.

Reviewer #1 (Remarks to the Author):

My previous comments were well-addressed by the authors in the new version. The authors have justified the novelty and clarified the underlying mechanism toward selective gold uptake. I, hence, glad to recommend its publication in NC in the present form.

Reply: We thank the reviewer's positive comment on our revised manuscript.

Reviewer #3 (Remarks to the Author):

Dear Authors,

Thank you for considering the comments and suggestions provided.

While considered point-by-point, these are not fully addressed from my point of view, some examples:

Reply: We thank the reviewer's comment, we apologize for not fully addressing all concerns listed previously. We will clarify these comments point-by-point in the following reply.

1. Concerning the amount of material, using a high amount of GO could be similar to using activated carbon. This is obviously easier and cheaper to produce. Although there is a large-scale production under development, the bulk-GO obtained could be different from the one reported in this study. Therefore, finding results similar to those reported here for large-scale water treatment is an ambitious bet.

Reply: We thank the reviewer's comment. Reviewer has concerned that, if the proposed rGO could be mass produced, its gold extraction behavior might significantly decrease compared to the results reported here, and would be similar to existing gold adsorbent-activated carbon. We feel, despite it does not affect the reported phenomena and mechanism study in this paper, this comment is a reasonable and general concern for nearly all laboratory-based researches, which aim for scaling up the laboratory results for the real applications, including ours.

However, as the large-scale production and commercialization are under development and not the focus of this manuscript, we believe it is too early to properly answer this comment. Nevertheless, with the evidence we have now in hand, we believe the results presented in this study will provide critical guidance and technical support to achieve similar performance when developing a large amount of rGO for real application to reclaim waste gold. Our rationales are:

- (1) We have used commercial GO as starting materials.
- (2) We have demonstrated, within our capability, the rGO and its membranes can be scaled up without sacrificing performance (Fig. 4a in the main text).
- (3) Furthermore, with the mechanism study, we identified the dispersibility and graphene areas of rGO to gold ions are two critical parameters for achieving high gold extraction capacity (Fig. 3f-h in main text).

To further address the reviewer's concern and make the readers aware this, we have added a short comment in the conclusion part main text, "though continuous efforts are required for its commercialization," before the statement of "rGO provides a considerable incentive for commercial recovery of gold from e-waste." This can be found in page 15 of the revised main text.

Regarding "Concerning the amount of material, using a high amount of GO could be similar to using activated carbon".

- (1) As there is no experimental evidence or published results to support this comment, we felt this might be better discussed when more evidence is presented during the future development.
- (2) Nevertheless, we have performed gold extraction by activated carbon derived from coconut shell (Henan JinLing Environmental Technology, used as purchased without further treatment), its gold extraction capacities (at room temperature) were ~300 mg/g and <1 mg/g for 10 ppm and 1 ppm $[\text{AuCl}_4]^-$, respectively. That is, rGO showed a much higher gold extraction capacity than the tested activated carbon.
- (3) The efficiency of gold extraction is defined by many factors beyond the extraction capacity. For example, as we have stated in the main text "introduction", activated carbon requires further elution and reduction of adsorbed gold ions. These processes are energy- and resource-intensive. Together with a relatively low gold extraction capacity, these pieces of evidence probably explain why there are so many researches on developing novel gold adsorbents as an alternative for activated carbon.

2. *The authors seem to be unclear about periodic calculations and molecular calculations (gas phase), the latter cannot be performed in VASP as discussed in the response letter. Although, in the revised manuscript, state-of-the-art density functional calculations are now correct, in the previous report, it was recommended to corroborate the theoretical predictions with molecular density calculations as a charged surface appears due to the presence of oxygen- functional groups (mainly hydroxyl groups) as well as the proposed Au system. For a charged system, the charge distribution is not correctly described by VASP or other related DFT software, see e.g., Physical Review Letters 126.7 (2021): 076401. Although this point is not mandatory, the authors should comment on the idealized and unrealistic system proposed by them.*

Reply: We thank the reviewer’s insightful comment. We have carefully read the recommended paper, and agree with reviewer that the conventional jellium charge correction implemented in VASP might give unrealistic one-electron energy and charge distribution. This can possibly result in that the obtained total Bader charge is idealized. Fortunately, the simplified scheme gives a consistent trend with experimental results, and more accurate results deserve further investigation either by self-consistent correction (Phys. Rev. Lett. 126, 076401 (2021)) or gas-phase simulations.

Following the reviewer’s suggestion, we have added the following comment on the idealized and unrealistic system used in our study to page 14 in SI:

“Here, we used the idealized charged systems, and a jellium background charge was added to ensure the whole system is neutral. It should be noted that, the jellium charge might introduce spurious states in the vacuum (Phys. Rev. Lett. 126, 076401 (2021)), which could result in the change to calculated one-electron energy and charge distribution and thus the Bader charge. Nevertheless, our DFT calculations showed a consistent trend with the experimental results. To obtain more accurate results, a self-consistent correction (Phys. Rev. Lett. 126, 076401 (2021)) or gas-phase simulations should be explored.”

3. *For the study of intraparticle diffusion, see: Scientific Reports 12.1 (2022): 1-12. How to calculate and discuss. The idea is to evidence the 2 or 3 regions of linearity through a simple linear fit, showing the adsorption behavior (from kinetic data) on the surface or internal structure. More importantly, the authors can properly discuss the intraparticle diffusion rate (k_p) and the intercept (C). The latter shows the boundary layer effect or surface adsorption. This point is not mandatory, but authors are encouraged to present this result for a better understanding of the adsorption process.*

Reply: We thank the reviewer’s comment. The intraparticle diffusion kinetic model (Sci. Rep. 12, 2534 (2022); Sci. Rep. 12, 6326 (2022)) can be written as:

$$q_t = k_p t^{1/2} + C$$

Where k_p and C are the intraparticle diffusion rate constant and the thickness of boundary layer, t is the adsorption time (h), and q_t is the amount of adsorbate on adsorbent at adsorption time of t , respectively.

Fig. R1 Intraparticle diffusion analysis on the kinetics of gold extraction by rGO.

(a) Plot of gold extraction capacity versus $t^{1/2}$. (b) Linear fit of the amount of gold extracted by rGO with an extraction time between 10 minutes to 24 h. The black dotted lines are the linear fits.

As discussed in the previous reply, the overall curve does not show linearity to $t^{1/2}$. While, following the reviewer's suggestion, the curve can be approximately segregated into three regions, and each region shows a linearity to $t^{1/2}$ (Fig. R1 a). The initial stage (stage 1) shows fast adsorption due to the availability of a large number of adsorption sites—graphene areas of rGO. We note, the sharp increase in adsorption capacity with time at this stage defines the rapid gold extraction as mentioned in Fig. 1c, and is in good agreement that rGO has abundant graphene areas. With the graphene areas gradually occupied by the gold ions/particles, the second adsorption stage (stage 2) shows gradual adsorption with decreased adsorption rates. The final adsorption is the equilibrium stage (stage 3) due to reduced availability of adsorption sites (graphene area) on rGO.

Furthermore, as the first stage happened rapidly within 10 min. For a proper analysis, we have analyzed the second adsorption stage which sufficient data points are presented (Fig. R1b). Fitting the linearity of the second stage, yields $y=178t^{1/2}+928$. That is, the intraparticle diffusion rate (k_p) is $178 \text{ mg}\cdot\text{g}^{-1}\cdot\text{h}^{-1/2}$ and the intercept (C) which defines the thickness of the boundary layer is 928.

For the k_p , k_p at stage 2 is smaller than that in stage 1, but is still significantly higher than the previous reported results (Sci. Rep. 12, 2534 (2022); Sci. Rep. 12, 6326 (2022); Sci. Rep. 12, 7362 (2022)), manifesting the fast and efficient gold extraction. Furthermore, the observed high intercept C is much higher than the reported data, suggesting the intraparticle diffusion is not the dominant rate-limiting step and a significant surface adsorption (Chem. Eng. J. 255, 97-106 (2014); Sci. Rep. 12, 2534 (2022); Sci. Rep. 12, 6326 (2022); Sci. Rep. 12, 7362 (2022); Chem. Eng. J. 350, 692-702 (2018)). These pieces of evidence are in agreement with our conclusion that the

reductive adsorption by the rGO surface dominated the gold adsorption behavior.

However, I recognize that the manuscript has improved remarkably, therefore I recommend its publication at the best criteria of the other Reviewers.

We thank the reviewer's acknowledgement on our revised manuscript.

Reviewer #4 (Remarks to the Author):

Authors revised very nicely the manuscript and it has been highly improved.

Reply: We thank the reviewer's positive comment. We are grateful that reviewer has found all your previous concerns are well clarified and addressed.

However, I did not know that similar phenomena were already published for MoS₂, when I reviewed before. I read the cited paper on MoS₂. Nature family journals should make much of the originality and creativity. The secondary study is less valued compared with the pioneering one. Therefore, I cannot recommend the publication.

Reply: We thank the reviewer's comment. We believe the reviewer's comment on "similar phenomena" refers to "reductive adsorption". Reviewer might also concern that both MoS₂ and rGO are 2D materials, so the previous report on MoS₂ influenced the novelty of this study. Our reply is as follows:

(1) The novelty and insights generated in this study have been clarified in the previous reply, and agreed by the reviewer 1 and 3, as the relevant reply does not appear in the previous reply to the reviewer, we would like to use the previous reply here to address the insights gained in this study.

“(1) We have identified a clear-cut fact that, >95% gold adsorption of rGO is reductive adsorption, this quantitative result is new and exciting. Previously reported adsorbents showed a mixed form of adsorption despite a reductive adsorption is involved. This allowed the ultrahigh extraction capacity to trace amount of gold as it maintained a low adsorbate (gold ion) concentration on the rGO, which has been discussed in detail at the start of our reply to reviewer.

(2) The predominant reduction of Au³⁺ to Au⁰ happens spontaneously, and supported the observed ultrahigh gold extraction capacity to trace amount of gold. In addition, this allowed us to reveal a site-specific adsorption for gold and other co-existing ions, that is, adsorption of gold ions mainly happened on the graphitic area of rGO, but adsorption of co-existing ions happened on the oxidized region of rGO by electrical charge based interaction. Such site-specific adsorption allows the precise selectivity of rGO, that is, when rGO is protonated, the functional groups lost their charge. Therefore, the previously adsorbed co-existing ions were stripped but did not affect the gold adsorption.

(3) Another insight is on, how to achieve an ultrahigh gold extraction capacity?

For the selection of gold ions, gold ions with reductive potential >0.8 eV can be reduced and extraction. For the selection of graphene adsorbent, not all graphene-based materials can efficiently extract gold (Fig 2). The graphene area of rGO reduces gold ions, and the oxidized region of rGO provides dispersibility that the graphene area can efficiently reduce gold ions. Both are important for a high extraction capacity.”

(2) To more specifically address the reviewer’s concern, we show here, there are distinct differences in gold adsorption performance, mechanism and post-processing method between MoS₂ and rGO. Specifically,

--Reductive adsorption is not new, indeed, this has been found in many novel gold adsorbents, even before the report on MoS₂, for example, Chem. Eng. J. 255, 97-106 (2014), it is a misunderstanding to summarize the novelties of this work as proposing a reductive adsorption mechanism. In our study, **we have identified a clear-cut fact that, >95% gold adsorption of rGO is reductive adsorption, this quantitative result is new and exciting.** Previously reported adsorbents, including MoS₂, showed a mixed form of adsorption despite a reductive adsorption is involved. This allowed the ultrahigh extraction capacity to trace amount of gold as it maintained a low adsorbate (gold ion) concentration on the rGO.

--The reductive mechanisms for rGO and MoS₂ are different. rGO relies on the redox reaction between the graphene area of rGO and gold ions, however, MoS₂ relies on the photocatalytic reduction of gold ions. Gold extraction by MoS₂ requires external light irradiation, however, rGO does not. This makes our process simpler and less demanding to gold extraction environment.

--The rGO showed ultrahigh gold extraction capacity, and this capacity remained high when the gold concentration in water is at trace amount. For example, MoS₂ showed >1000 mg/g extraction capacity at a gold concentration of hundreds ppm, but this is still lower than rGO (Table S1). Furthermore, at an e-waste recycling relevant concentration, for example, tens of ppm or lower, MoS₂ showed <50 mg/g gold extraction capacity to 10 ppm gold, however, the rGO has a capacity of 1850 mg/g at the same concentration. These comparison has been listed in Table S1.

--Post-processing. Separation of MoS₂ and adsorbed gold involves dissolving gold and reduction (ACS EST Engg. 1, 1342-1350 (2021)), which is similar to the process of activated carbon, and would be energy- and resource-intensive.

These key differences make rGO distinct from MoS₂, we therefore, believe the novelty of our work is not undermined by the previous reports on MoS₂.